# The exocytosis regulator complexin controls spontaneous synaptic vesicle release in a CAPS-dependent manner at *C. elegans* excitatory synapses

Ya Wang[1☯], Chun Hin Chow[2,3☯], Yu Zhang[1], Mengjia Huang[2,3], Randa Higazy[2,3,4], Neeraja Ramakrishnan[4], Lili Chen[1], Xuhui Chen[5], Yixiang Deng[1], Sheng Wang[1], Cuntai Zhang[5], Cong Ma[1]*, Shuzo Sugita[2,3]*, Shangbang Gao [1,5‡]*

1 Key Laboratory of Molecular Biophysics of the Ministry of Education, College of Life Science and Technology, Huazhong University of Science and Technology, Wuhan, China, 2 Division of Experimental & Translational Neuroscience, Krembil Brain Institute, University Health Network, Ontario, Canada, 3 Department of Physiology, Temerty Faculty of Medicine, University of Toronto, Toronto, Ontario, Canada, 4 Lunenfeld-Tanenbaum Research Institute, Mount Sinai Hospital, University of Toronto, Toronto, Ontario, Canada, 5 Key Laboratory of Vascular Aging of the Ministry of Education, Tongji Hospital of Tongji Medical College, Huazhong University of Science and Technology, Wuhan, China

☯ These authors contributed equally to this work.
‡ Lead contact.
* cong.ma@hust.edu.cn (CM); Shuzo.Sugita@uhnresearch.ca (SS); sgao@hust.edu.cn (SG)

**Data Availability Statement:** Original, uncropped, and minimally adjusted images used to generate the graphs in Fig 5 can be accessed in the 'S1 Raw

## Abstract

The balance between synaptic excitation and inhibition (E/I) is essential for coordinating motor behavior, yet the differential roles of exocytosis regulators in this balance are less understood. In this study, we investigated the roles of 2 conserved exocytosis regulators, complexin/CPX-1 and CAPS/UNC-31, in excitatory versus inhibitory synapses at *Caenorhabditis elegans* neuromuscular junctions. *cpx-1* null mutants exhibited a marked increase in spontaneous release specifically at excitatory synapses, alongside an unequal reduction in excitatory and inhibitory evoked release. A clamping-specific knockin mutant, *cpx-1(Δ12)*, which preserved evoked release, also showed a biased enhancement in excitatory spontaneous release. Conversely, the *unc-31* null mutation, while maintaining normal spontaneous release, displayed a more pronounced reduction in evoked release at excitatory synapses. Notably, we found that CPX-1's clamping function is dependent on UNC-31 and is sensitive to external $Ca^{2+}$. Pull-down experiments confirmed that CAPS/UNC-31 does not directly interact with complexin, implying an indirect regulatory mechanism. Moreover, complexin regulates activity-dependent synaptic plasticity, which is also UNC-31 dependent. The unexpected role of CAPS/UNC-31 in the absence of CPX-1 clamping function may underpin the synaptic E/I balance and coordinated behavioral outputs in different species.

Images' folder within the Supplementary Information files.

**Funding:** This research was supported by the Major International (Regional) Joint Research Project (32020103007 to S.G., https://www.nsfc.gov.cn/), the National Key Research and Development Program of China (2022YFA1206001 to S.G., https://www.most.gov.cn/), the National Natural Science Foundation of China (32371189 to S.G.; 32300984 to L.C., https://www.nsfc.gov.cn/), the Natural Sciences and Engineering Research Council of Canada (RGPIN 2020 07139 to S.S., https://www.nserc-crsng.gc.ca/) and the Canadian Institute of Health Research (CIHR PJT 165917 to S.S., https://cihr-irsc.gc.ca/). The funders had no role in study design, data collection and analysis, decision to publish, or preparation of the manuscript.

**Competing interests:** The authors have declared that no competing interests exist.

**Abbreviations:** AD, accessory domain; CH, central α-helix; CTD, C-terminal domain; DCV, dense-core vesicle; eEPSC, evoked excitatory postsynaptic current; E/I, excitation and inhibition; eIPSC, evoked inhibitory postsynaptic current; GST, glutathione S-transferase; KI, knock-in; MHD1, Munc13 homology domain-1; mIPSC, miniature inhibitory postsynaptic current; mPSC, miniature postsynaptic current; NGM, nematode growth medium; NMJ, neuromuscular junction; NTD, N-terminal domain; RRP, readily releasable pool; SNARE, Soluble N–ethylmaleimide sensitive factor Attachment protein Receptor; SV, synaptic vesicle; VGCC, voltage-gated calcium channel.

## Introduction

Maintaining excitatory and inhibitory (E/I) balance is a fundamental property of neuronal circuits. For example, the balance between excitatory and inhibitory synaptic transmission in the motor circuitry is critical for generating coordinated locomotion. Altered E/I balance in the brain underlies epilepsy and is implicated in psychiatric disorders such as autism [1–3]. Modulation of the synaptic vesicular release of neurotransmitters is an important mechanism for controlling the relative strength and weight of excitatory and inhibitory inputs [4]. In both types of synapses, the neuronal Soluble N–ethylmaleimide sensitive factor Attachment protein REceptor (SNARE) proteins drive the fusion of neurotransmitter-filled synaptic vesicles with the plasma membrane [5]. Much of our previous understanding of the regulation of SNARE-mediated vesicular fusion comes from mammalian glutamatergic excitatory synapses, leaving the regulatory mechanisms in inhibitory synapses less explored [6]. In *Caenorhabditis elegans*, at least 11% of functional neuronal connections are inhibitory [7]; however, the differences in the regulation between excitatory and inhibitory synapses are largely unknown.

The highly regulated and synchronous process of vesicular fusion at the synapse requires an array of regulatory proteins at the presynaptic terminal. In most synapses, a pool of primed (i.e., readily releasable) vesicles is maintained until synaptotagmin senses $Ca^{2+}$ influx and triggers release [8,9]. Complexin is a key synaptic vesicle secretory regulatory protein that functionally and structurally interacts with synaptotagmin to keep vesicles in a primed state before evoked release [10–12]. Complexins are evolutionarily conserved proteins composed of N-terminal and C-terminal regions with a central α-helix (CH) and an accessory domain (AD) [13,14]. Studies on mouse, *Drosophila*, and *C. elegans* complexin null mutants have shown consistent reductions in synchronous $Ca^{2+}$-dependent evoked excitatory exocytosis [15–22]. In particular, the N-terminal domain is critical for promoting fast fusion events underlying evoked release [23,24]. On the other hand, the C-terminal domain (CTD) of complexin is proposed to perform the inhibitory clamping function that "clamps" synaptic vesicles to the primed state [12,23–27]. The clamping function of complexin is evident in which spontaneous neurotransmitter release is enhanced when complexin expression is decreased or fully abolished [21,22,28–30]. However, a few mammalian studies have shown reduced spontaneous release in the absence of complexin, making the clamping functions of complexin controversial [15–17,20]. Furthermore, in mouse cortical neurons, complexin was shown to clamp spontaneous exocytosis by blocking a secondary $Ca^{2+}$ sensor, in addition to synaptotagmin-1 [31]. However, the presynaptic partners that interact with complexin remain unknown. Regardless, the dual functions of complexin are formulated on the basis of studies utilizing excitatory synapses. In inhibitory synapses, removing all 3 mammalian complexin isoforms in cultured GABAergic striatal neurons reportedly decreases both evoked and spontaneous release [15], in contrast with the dual regulation in many excitatory synapses. In *C. elegans*, the expression of complexin in excitatory acetylcholine motor neurons, but not in inhibitory GABAergic motor neurons, rescued aldicarb hypersensitivity in *cpx-1* mutants [19,22]. The differential impact of altering complexin in excitatory versus inhibitory synapses strongly suggests that the roles of complexin need to be independently studied at each synapse.

CAPS is another crucial synaptic vesicle secretory regulatory protein [32] that is recognized primarily for its ability to modulate dense-core vesicle (DCV) exocytosis [33–39]. In contrast to the role of Munc13/UNC-13, which functions as a critical priming factor, the involvement of CAPS/UNC-31 in regulating synaptic vesicle exocytosis has been a topic of extensive debate. As an evolutionarily conserved calcium-binding protein, CAPS/UNC-31 contains 2 major domains: a central PH domain that binds to phospholipids and a C-terminal domain that is homologous to the Munc13 protein domain involved in synaptic vesicle priming. Null

mutants for homologs of CAPS/UNC-31 in *C. elegans* (*unc-31*) and *Drosophila* (*dCAPS*) show a loss of secretion of neurotransmitters and neuropeptides [40,41]. Speese and colleagues further reported that *unc-31* null *C. elegans* mutants exhibited reduced peptide release in vivo but had normal stimulated synaptic vesicle recycling in cultured neurons [42]. With a domain homologous to that of Munc13, CAPS/UNC-31 is proposed to dock DCVs [43,44]. However, several lines of evidence suggest that synaptic transmission also, at least in part, depends on CAPS/UNC-31. Renden and colleagues demonstrated that *Drosophila dCAPS* mutants exhibit mild reductions in excitatory synaptic transmission [45]. CAPS1 and CAPS1&2 double knock-out mice exhibit a strong decrease in synaptic vesicle exocytosis at excitatory synapses [46]. Finally, the loss of UNC-31 in *C. elegans* also results in decreased evoked excitatory transmission [47,48]. Nevertheless, a reduction in synaptic transmission caused by the deletion of CAPS/UNC-31 is generally considered secondary to a reduction in DCV release in the *C. elegans* field [42]. Several questions regarding CAPS/UNC-31 are crucial: (1) Is there a differential regulatory effect of CAPS/UNC-31 on excitatory versus inhibitory synapses? (2) Does CAPS/UNC-31 directly control the release of synaptic vesicles independent of its function in DCV docking? (3) How does CAPS/UNC-31 functionally interact with other synaptic regulatory proteins, such as complexin?

In this study, we investigated the roles of complexin/CPX-1 and CAPS/UNC-31 and their functional dependence in synaptic vesicle exocytosis at both excitatory and inhibitory synapses at *C. elegans* neuromuscular junction. Our results revealed that while the *cpx-1* null mutation decreased evoked release in both excitatory and inhibitory synapses, it enhanced spontaneous release only in excitatory synapses. We showed that CAPS/UNC-31 played a significant role in evoked, but not spontaneous, synaptic vesicle exocytosis independent of dense-core vesicle regulation. Deficits in evoked exocytosis of *unc-31* were stronger in excitatory synapses than in inhibitory synapses. Intriguingly, when spontaneous release was enhanced by the loss of complexin/CPX-1 clamping functions, CAPS/UNC-31 became indispensable for the maintenance of elevated spontaneous exocytosis in the presence of extracellular calcium. Our results demonstrated the differential regulation of excitatory versus inhibitory synaptic vesicle exocytosis by complexin/CPX-1 and CAPS/UNC-31. The requirement of UNC-31 in the absence of CPX-1 clamping function may underlie the basis of the synaptic E/I balance and complex behavioral phenotypes of *cpx-1* and *unc-31* mutants.

## Results

### *cpx-1* mutants display impaired locomotion

The *C. elegans* genome contains 2 complexin genes: *cpx-1* and *cpx-2*. *cpx-1* encodes the CPX-1 protein, which has a relatively high degree of homology to mammalian complexins and is considered the major complexin functioning in the nervous system in *C. elegans* [19,22]. The CPX-1 protein consists of four functional domains, the N-terminal domain (NTD), accessory domain (AD), central α-helix (CH), and C-terminal domain (CTD) (**Fig 1A**), each of which play distinct functional roles [16,19,23,26,49–52]. To examine the contribution of CPX-1, we investigated motor behavioral defects caused by *cpx-1* mutations via various assays.

We first found that a null mutant allele, *cpx-1 (ok1552)*, which contains a large fragment deletion of sequences encoding the N-terminal domain, accessory domain, central α-helix, and part of the C-terminal domain (**Figs 1A and S1A**), showed a reduced ability to coordinate body oscillations during thrashing assays (**Fig 1B**). Next, we used an aldicarb-induced paralysis assay to evaluate whether the synaptic transmission of *cpx-1* mutants is altered. Aldicarb is an acetylcholinesterase inhibitor that prevents the removal of acetylcholine at the synapse [53,54]. Increased aldicarb sensitivity is commonly associated with heightened excitatory transmission

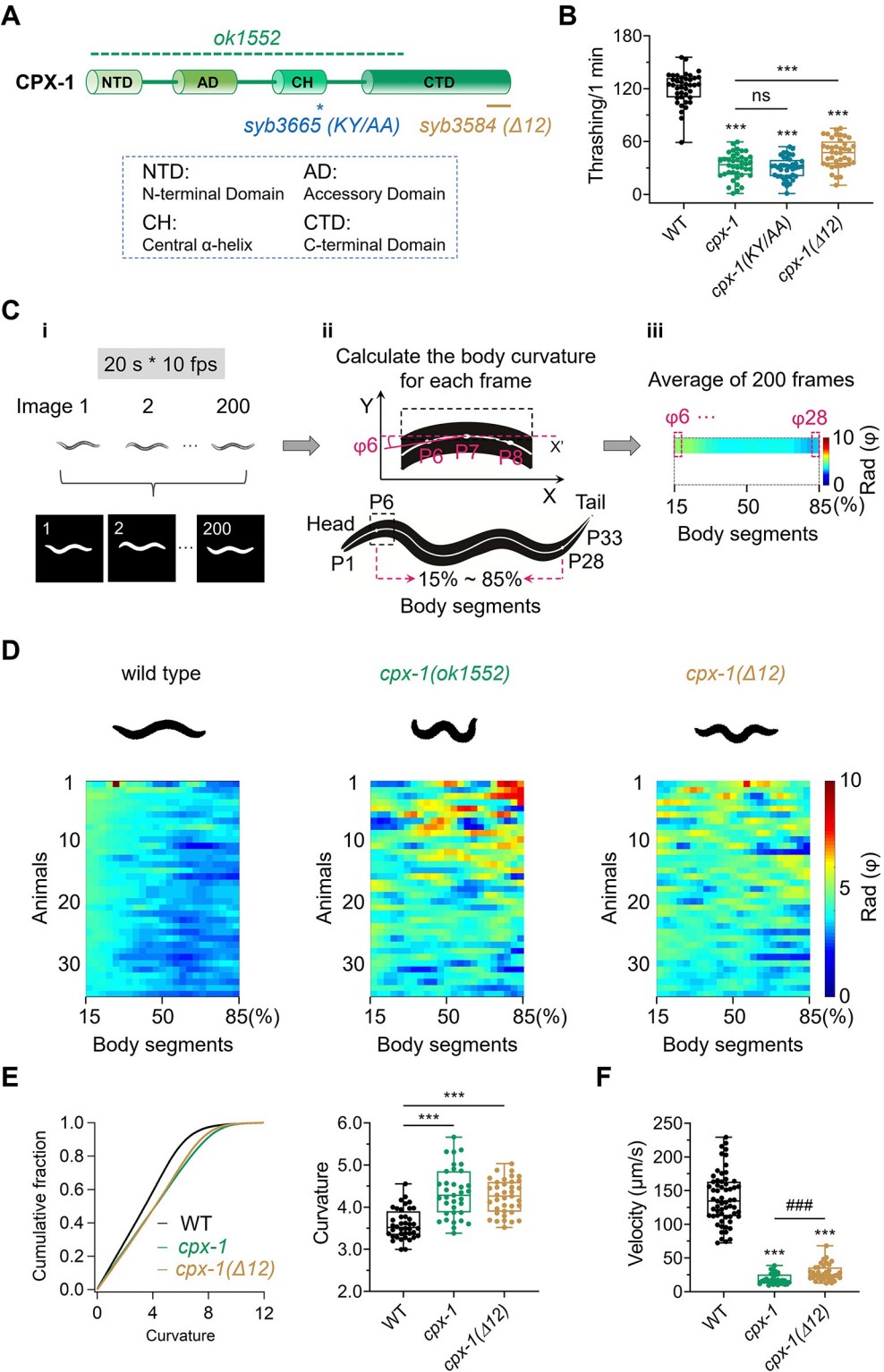

**Fig 1. *cpx-1* mutants display defective locomotion and increased body curvature.** (**A**) Cartoon representation of the structure of CPX-1, indicating the NTD, AD, CH, and CTD. Mutation sites of *ok1552* (green line), *syb3665* (blue stars), and *syb3584* (brown line) are labeled. (**B**) Motility measured by thrashing number per minute across the wild type, *cpx-1(ok1552)*, *cpx-1(KY/AA, syb3665)*, and *cpx-1(Δ12, syb3584)*. *n* ≥ 40 animals. One-way ANOVA was performed ($F_{(3, 156)} = 308.1$, $P < 0.0001$). (**C**) Schematic representation of the method used to measure body curvature. (i) Each

free-moving worm was imaged for 20 s at 10 Hz, and 200 images showing the continuous locomotion of the worm were obtained. (ii) The worm body in each frame was divided into 32 body segments, and the curvature was calculated via MATLAB. The curvatures of the body segments at 15%–85% ($\phi6$—$\phi28$) were used for subsequent analysis. (iii) The curvature obtained in (ii) is plotted as a color map. Each pixel represents the average of 200 images of the curvature of a body segment. (**D**) Representative body bending states (top) and curvature (bottom) in the indicated genotypes. (**E**) Distribution and quantification of body curvature at the indicated strains. Each point represents the average of the 15%–85% curvature of the body segments for 20-s-locomotion for each worm. $n \geq 35$ animals. One-way ANOVA was performed ($F_{(2, 109)} = 30.66$, $P < 0.0001$). (**F**) Quantification of locomotion velocities in wild-type, *cpx-1(ok1552)*, and *cpx-1(Δ12, syb3584)* worms. $n \geq 33$ animals. One-way ANOVA was performed ($F_{(2, 126)} = 285.5$, $P < 0.0001$). Data are presented as box-and-whisker plots, with the median (central line), 25th–75th percentile (bounds of the box), and the 5th–95th percentiles (whiskers) are indicated in (**B**, **E**, **F**). Student's *t* test was performed for comparisons of 2 groups, ### $P < 0.001$. One-way ANOVA was used for comparisons of multiple groups, followed by Tukey's range test, *** $P < 0.001$; ns, not significant. The error bars represent the standard error of the mean (SEM). $N = 3$ independent replicates. All the raw data associated with this figure are available in S1 Data. AD, accessory domain; CH, central α-helix; CTD, C-terminal domain; NTD, N-terminal domain.

or diminished inhibitory transmission in *C. elegans*, fundamentally stemming from a disruption in the E/I balance [54,55]. Compared with wild-type N2 animals, the *cpx-1(ok1552)* mutant exhibited significantly greater aldicarb sensitivity (**S1B**). These findings suggest that motor behavior is impaired in the absence of CPX-1.

To further explore the function of CPX-1, we generated 2 additional knock-in (KI) mutants via CRISPR/Cas9 via the proposed essential domains. The first mutation involves the deletion of the C-terminal 12 residues (*Δ12, syb3584*) (**S1A Fig**), which are suggested to be critical for its clamping function in suppressing spontaneous release [23–26]. The second mutant, KY/AA (*syb3665*), targets the central α-helix (**S1A Fig**), which is a key domain for SNARE binding [19,30,49,56]. We compared the thrashing and aldicarb phenotypes of these 2 KI mutants with those of wild-type and *cpx-1(ok1552)* worms. *cpx-1(Δ12, syb3584)* exhibited significant but mild thrashing defects and heightened aldicarb sensitivity (**Figs 1B and S1B**). Conversely, *cpx-1(KY/AA, syb3665)* worms presented thrashing defects and aldicarb sensitivity identical to those of *cpx-1(ok1552)* mutants (**Figs 1B and S1B**). These behavioral defects further demonstrate that CPX-1 is essential for regulating locomotion coordination.

Additionally, *cpx-1 (ok1552)* worms presented a body bending defect (**S1 and S2 Movies**). Quantitative analysis of the average body curvature in the free-moving worms (**Fig 1C**) revealed that the *cpx-1(ok1552)* mutant presented a hypercontraction muscle phenotype, similar to that observed in the aldicarb assay and when cholinergic motor neurons were overactivated [54,57]. To exclude errors due to unstable head and tail oscillations, the curvature was calculated for 15% to 85% of the body segments (**Fig 1Cii**). A significant increase in body curvature was also observed in *cpx-1(Δ12)* worms (**Fig 1D and 1E**). Consequently, increased body bending was accompanied by severe locomotion defects, with a significant reduction in the speed of movement (**Fig 1F**). These results demonstrate that CPX-1 regulates locomotion coordination and suggest that there is an underlying imbalance between excitatory and inhibitory synaptic transmission in *cpx-1* mutants.

## *cpx-1* mutations lead to an E/I imbalance in synaptic transmission

Most electrophysiological analyses of synaptic transmission in *C. elegans* have been limited to excitatory transmission [19,22,47,48], leaving the mechanism of inhibitory synaptic release less well understood. We examined the roles of complexin/CPX-1 in both excitatory and inhibitory neuromuscular junctions (NMJs) in *C. elegans*. Both spontaneous release and evoked release were measured. We used *cpx-1* mutants with *zxIs6* (excitatory synapse) or *zxIs3* (inhibitory synapse) lines to measure exocytosis via optogenetics [57]. The selective expression of

channelrhodopsin in excitatory cholinergic or inhibitory GABAergic neurons enabled reliable measurement of exocytosis underlying neurotransmission.

*cpx-1* null mutants reportedly exhibit enhanced spontaneous release and reduced evoked release from excitatory motor neurons [19,22]. Compared with wild-type animals, *cpx-1 (ok1552)* increased the frequency and amplitude of miniature postsynaptic currents (mPSCs) (**Fig 2A–2C**). Conversely, the evoked excitatory postsynaptic current (eEPSC) recorded in *cpx-1(ok1552)* mutants was drastically lower than that recorded in wild-type animals (**Fig 2D–2F**). Thus, *C. elegans* CPX-1 aligns with the widely accepted perspective of its dual functions—facilitating evoked release while inhibiting spontaneous release at excitatory synapses.

Given that *C. elegans* NMJs receive both cholinergic and GABAergic inputs simultaneously, mixed excitatory (predominantly) and inhibitory (minor) mPSCs were collected under our recording conditions [58]. To isolate GABAergic inhibitory transmission in *cpx-1(ok1552)*, we employed D-tubocurarine (d-TBC, 0.5 mM), a broad-spectrum blocker of *C. elegans* ionotropic AChRs, to isolate miniature inhibitory postsynaptic currents (mIPSCs) [58,59]. Interestingly, compared with those in wild-type worms, neither the frequency nor the amplitude of mIPSCs was altered in *cpx-1(ok1552)* mutants (**Fig 2G–2I**). Thus, the increased mPSCs observed in the *cpx-1* mutant could be attributed to its function in excitatory synapses. Nevertheless, evoked inhibitory postsynaptic currents (eIPSCs) were decreased in the *cpx-1(ok1552)* mutant (**Fig 2J–2L**). Therefore, unlike excitatory NMJs, where a lack of CPX-1 results in decreased evoked release coupled with enhanced spontaneous release, inhibitory synapses do not exhibit this coupling.

To delve deeper into the distinct impacts of *cpx-1(ok1552)* on excitatory and inhibitory transmission, we conducted a comparative analysis of its preferential influence on the 2 synapses. While mIPSCs were unaltered in *cpx-1(ok1552)* mutant animals, the frequency of mPSCs increased, indicating that spontaneous release at the NMJ without CPX-1 was driven to a heightened excitatory state (**Fig 2M**, spontaneous release). For evoked release, we analyzed the relative magnitude changes ($\Delta = I_{WT}/I_{cpx-1}$) and found that the peak amplitude of eEPSC decreased by a factor of $\Delta = 11.56$, whereas the reduction in eIPSC was only $\Delta = 3.04$ (**Fig 2E and 2K**) in *cpx-1(ok1552)* mutants. This yielded a notable 3.8-fold disparity in the decrease in excitatory evoked transmission compared with inhibition. A similar trend was observed when we compared the relative change in transferred charges, which showed a 4.1-fold disparity between eEPSCs ($\Delta = 6.57$) and eIPSCs ($\Delta = 1.59$) (**Fig 2F and 2L**). Therefore, evoked transmission in *cpx-1(ok1552)* mutants preferentially weakened excitatory strength compared with inhibition (**Fig 2M**, evoked release).

Collectively, these findings indicate that *cpx-1* mutation enhances the frequency and amplitude of miniature PSCs but not inhibitory miniature PSCs. These findings imply that complexin/CPX-1 had a more pronounced influence on excitatory synapses than on inhibitory synapses (E > I) at the neuromuscular junction. Thus, *cpx-1* mutations lead to an imbalance between excitatory and inhibitory synaptic transmission, which in turn leads to motor behavioral deficits (**Fig 2M**). Considering that most components of the release machinery, including complexin, participate in both excitatory and inhibitory synapses, the predominant function of CPX-1 in excitatory synapses rather than inhibitory synapses suggests a coordinated or balanced mechanism between CPX-1 and other release machinery (see Discussion).

## Clamping-specific mutant *cpx-1(Δ12)* prominently enhances spontaneous excitatory release but does not affect evoked release

The decrease in excitatory synchronous release may have been due to the increase in spontaneous release depleting the primed vesicles [60]. However, the opposite regulation by CPX-1 of

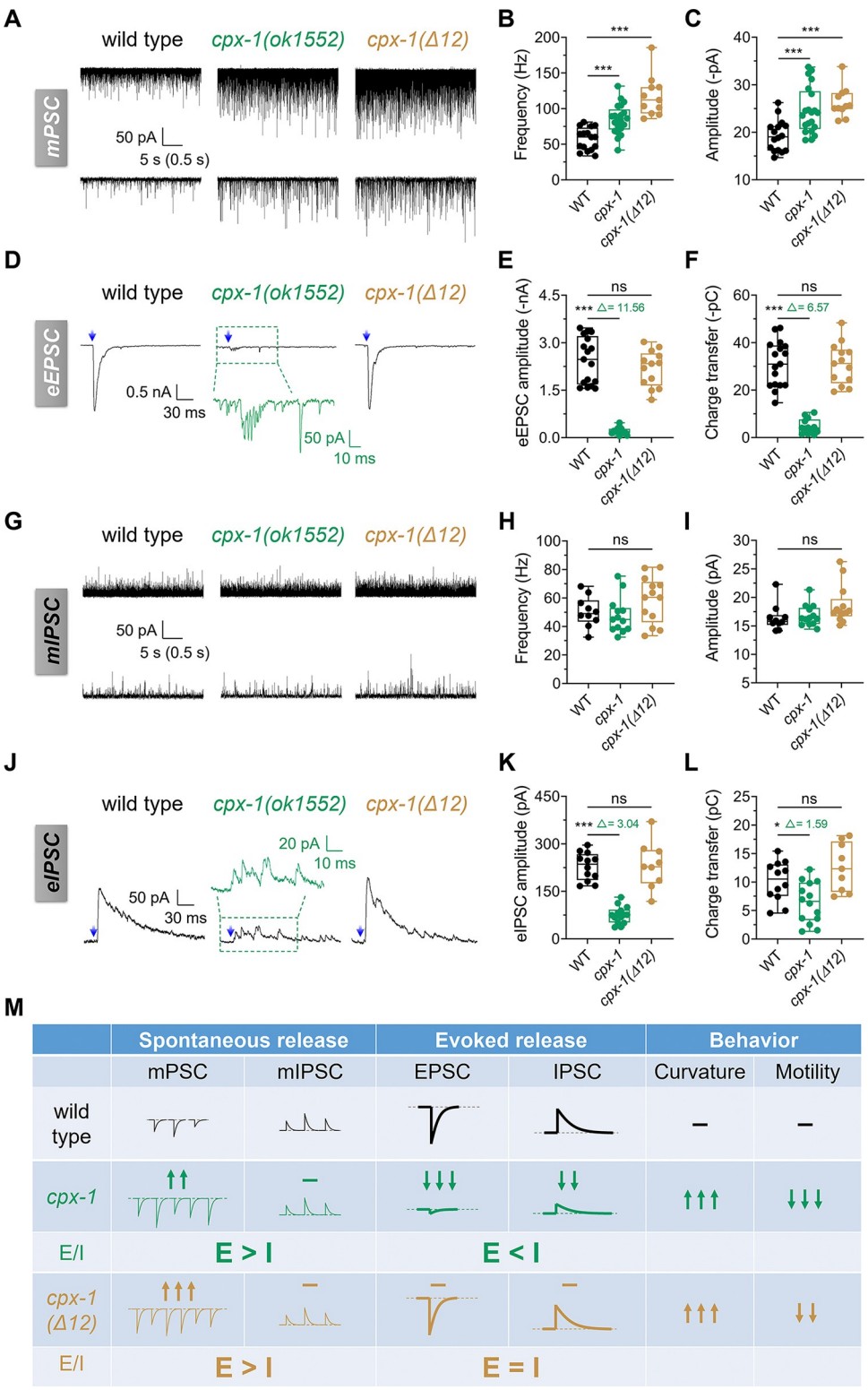

**Fig 2. *cpx-1* mutations lead to an imbalance between excitatory and inhibitory synaptic transmission. (A)** Representative traces of mPSCs recorded from wild-type, *cpx-1(ok1552)*, and *cpx-1(Δ12)* worms at different timescales. Recordings were made from body wall muscle cells at −60 mV. **(B, C)** Quantification of the mPSC frequency **(B)** and amplitude **(C)** in each genotype. $n \geq 11$ animals. One-way ANOVA was performed ($F_{(2, 45)} = 26.01$, $P < 0.0001$ for **B**; $F_{(2, 45)} = 11.94$, $P < 0.0001$ for **C**). **(D)** Representative traces of eEPSCs recorded from the indicated genotypes in the

*zxIs6* [P*unc-17*::ChR2::YFP + *lin-15*(+)] transgenic background with 10 ms blue light illumination (blue arrow). The lower trace of *cpx-1(ok1552)* (green) shows a zoomed-in region from the upper trace. (**E**, **F**) Quantification of eEPSC amplitude (**E**) and charge transfer (**F**) at different strains. The numbers after Δ represent the ratio of the average values of WT and *cpx-1 (ok1552)*. $n \geq 13$ animals. One-way ANOVA was performed ($F_{(2, 40)} = 65.01$, $P < 0.0001$ for **E**; $F_{(2, 40)} = 50.07$, $P < 0.0001$ for **F**). (**G**) Representative traces of mIPSCs recorded from wild-type, *cpx-1(ok1552)*, and *cpx-1 (Δ12)* worms. The muscles were held at −10 mV with the 0.5 mM ionotropic acetylcholine receptor blocker D-tubocurarine (d-TBC). (**H**, **I**) Quantification of the mIPSC frequency (**H**) and amplitude (**I**) in each genotype. $n \geq 10$ animals. One-way ANOVA was performed ($F_{(2, 33)} = 2.313$, $P = 0.1148$ for **H**; $F_{(2, 33)} = 2.442$, $P = 0.1026$ for **I**). (**J**) Representative traces of inhibitory postsynaptic currents (eIPSCs) recorded from the indicated strains in the *zxIs3* [P*unc-47*::ChR2::YFP + *lin-15*(+)] transgenic background with 10 ms blue light illumination (blue arrow). (**K**, **L**) Quantification of the eIPSC amplitude (**K**) and charge transfer (**L**) at the indicated strains. The numbers after Δ represent the ratio of the average values of WT and *cpx-1 (ok1552)*. $n \geq 9$ animals. One-way ANOVA was performed ($F_{(2, 32)} = 43.37$, $P < 0.0001$ for **K**; $F_{(2, 32)} = 8.225$, $P = 0.0013$ for **L**). (**M**) Schematic representation illustrating the imbalance of excitatory and inhibitory synaptic transmission due to *cpx-1* mutations and resulting behavioral defects. The data are presented as box-and-whisker plots, with the median (central line), 25th–75th percentile (bounds of the box), and 5th–95th percentile (whiskers) indicated. One-way ANOVA was used for comparisons of multiple groups, followed by Tukey's range test, * $P < 0.05$; *** $P < 0.001$; ns, not significant. The error bars represent the SEM. $N = 3$ independent replicates. All the raw data associated with this figure are available in S1 Data. eEPSC, evoked excitatory postsynaptic current; eIPSC, evoked inhibitory postsynaptic current; mIPSC, miniature inhibitory postsynaptic current; mPSC, miniature postsynaptic current.

spontaneous and evoked release indicates that its functional domain mechanism should be segregated [19,22,61]. To date, no clamping-specific mutant that preserves evoked release has been identified in *C. elegans*. Therefore, we aimed to generate such a mutant in this study. Compared with the *cpx-1(ok1552)* mutant, the *cpx-1(Δ12)* mutant presented better motility (**Fig 1B** and **S2 and S3 Movies**) and heightened aldicarb sensitivity (**S1B Fig**); both phenotypes are consistent with genetic lesions selectively disrupting the clamping ability of CPX-1 without altering evoked synaptic transmission.

By measuring synaptic transmission via electrophysiology, we found that *cpx-1(Δ12)* retained normal evoked release in both excitatory (**Fig 2D–2F**) and inhibitory synapses (**Fig 2J–2L**). As expected with high aldicarb sensitivity, *cpx-1(Δ12)* strongly increased the frequency and amplitude of mPSCs (**Fig 2A–2C**). Similar to *cpx-1(ok1552)* worms, *cpx-1(Δ12)* mutants did not alter the frequency or amplitude of mIPSCs (**Fig 2G–2I**). Given that evoked release was normal in *cpx-1(Δ12)*, we demonstrated that evoked and spontaneous release can be segregated in complexin/CPX-1, where increased spontaneous activity does not necessarily deplete primed vesicles for synchronous release.

Furthermore, *cpx-1(Δ12)* worms specifically promoted spontaneous release in excitatory synapses without affecting inhibitory neurotransmission, leading to an E-I imbalance that dominated excitatory transmission (**Fig 2M**). In summary, CPX-1 exerted a stronger modulatory effect on excitatory synapses than on inhibitory synapses in *C. elegans*.

## CAPS/UNC-31 differentially regulates evoked excitatory and inhibitory neurotransmission

We further investigated other regulatory partners in these 2 synapses. Prominent proteins in the presynaptic secretion machinery, such as Munc13, Munc18, and Synaptotagmins, have been extensively studied [27,62–64]. However, the impact of *unc-31* mutants on excitatory evoked release has yielded contradictory results [42,46–48], and their phenotype in inhibitory synapses has not been thoroughly investigated. Here, we examined the *unc-31(e928)*-null mutant to evaluate the role of UNC-31 in E/I balance regulation.

First, we assessed the body curvature of *unc-31(e928)*-null mutants and found that it was significantly lower than that of the wild type (**Fig 3A–3C** and **S1 and S4 Movies**). The *unc-31* mutant animals exhibited a rigid and straight posture, suggesting excessive muscle relaxation,

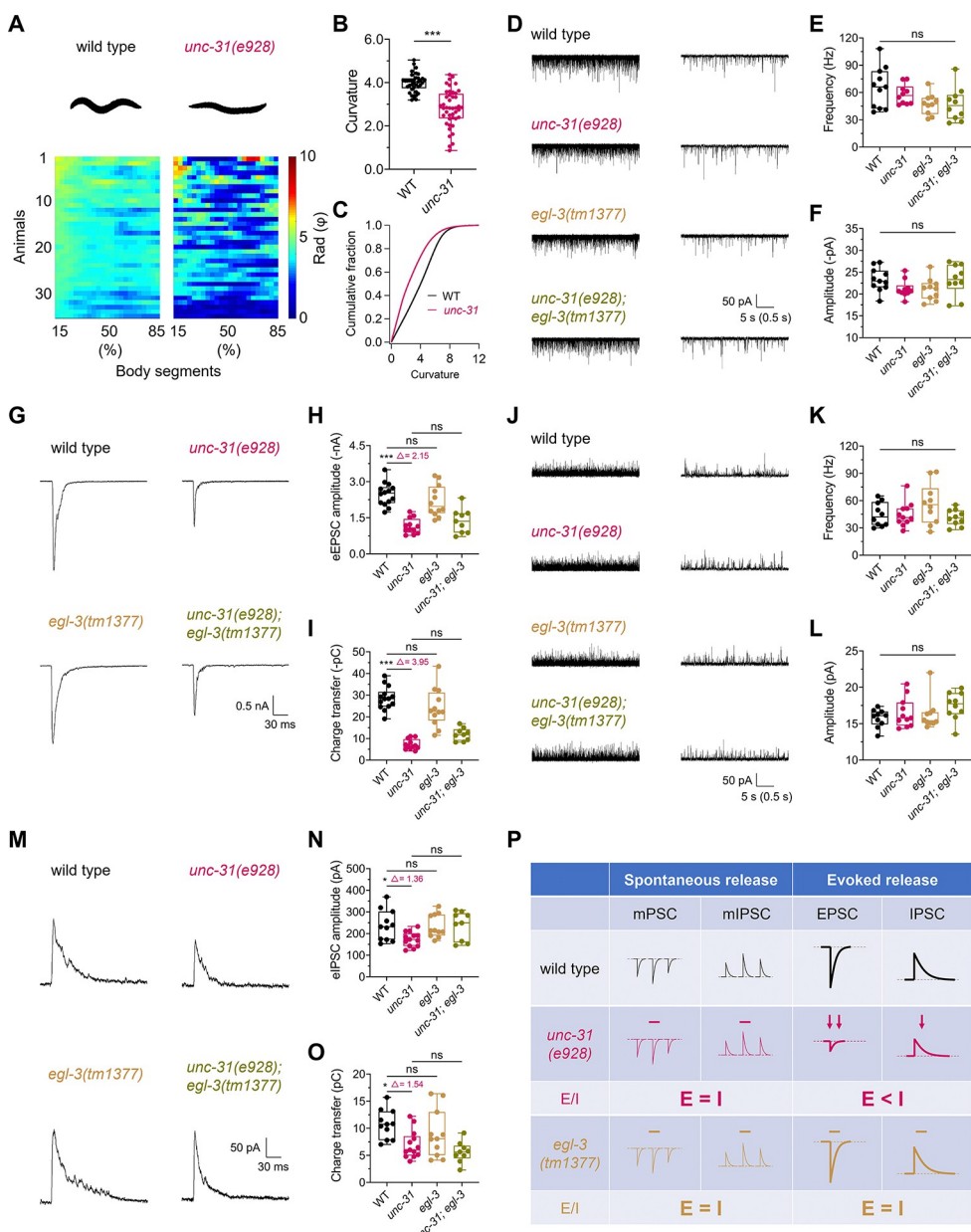

**Fig 3. UNC-31/CAPS differentially regulates evoked cholinergic and GABAergic neurotransmission.** (**A**) Representative body bending states (top) and curvature (bottom) in wild-type and *unc-31(e928)* animals. (**B, C**) Quantification (**B**) and distribution (**C**) of body curvature in WT and *unc-31* mutant animals. $n \geq 37$ animals. (**D**) Representative traces of mPSCs recorded from WT, *unc-31(e928)*, *egl-3(tm1377)*, and *unc-31(e928); egl-3(tm1377)* worms at different timescales. (**E, F**) Quantification of the mPSC frequency (**E**) and amplitude (**F**) across different genotypes. $n \geq 10$ animals. One-way ANOVA was performed ($F_{(3, 37)}$ = 3.028, $P$ = 0.0415 for **E**; $F_{(3, 37)}$ = 2.065, $P$ = 0.1216 for **F**). (**G–I**) Representative traces and quantification of evoked EPSCs recorded from the indicated strains. The numbers after $\Delta$ represent the ratio of the average values of WT and *unc-31 (e928)*. $n \geq 9$ animals. One-way ANOVA was performed ($F_{(3, 43)}$ = 19.95, $P < 0.0001$ for **H**; $F_{(3, 43)}$ = 39.33, $P < 0.0001$ for **I**). (**J**) Representative traces of mIPSCs recorded from the indicated strains are shown at different timescales. (**K, L**) Quantification of mIPSC frequency (**K**) and amplitude (**L**) among different genotypes. $n \geq 10$ animals. One-way ANOVA was performed ($F_{(3, 37)}$ = 1.896, $P$ = 0.1472 for **K**; $F_{(3, 37)}$ = 1.466, $P$ = 0.2395 for **L**). (**M–O**) Representative traces and quantification of evoked IPSCs recorded from the indicated strains. The numbers after $\Delta$ represent the ratio of the average values of WT and *unc-31 (e928)*. $n \geq 9$ animals. One-way ANOVA was performed ($F_{(3, 40)}$ = 3.709, $P$ = 0.0191 for **N**; $F_{(3, 40)}$ = 5.71, $P$ = 0.0024 for **O**). (**P**) Schematic representation illustrating the imbalance of excitatory and inhibitory synaptic transmission in the *unc-31* mutant. The data are presented as box-and-whisker plots, with the median (central line), 25th–75th percentile (bounds of the box), and 5th–95th percentile (whiskers) indicated. Student's *t* test was performed for comparisons of 2 groups, and one-way ANOVA was used for comparisons of multiple groups. Tukey's range test

was subsequently performed, * $P < 0.05$; *** $P < 0.001$; ns, not significant. The error bars represent the SEM. $N = 3$ independent replicates. All the raw data associated with this figure are available in S1 Data. mIPSC, miniature inhibitory postsynaptic current; mPSC, miniature postsynaptic current.

in contrast to the *cpx-1* mutants (S2 and S4 Movies). Electrophysiological analyses revealed that *unc-31(e928)* did not impact spontaneous release in either excitatory or inhibitory synapses, as evidenced by the unchanged frequency and amplitude of mPSCs and mIPSCs (Fig 3D–3F and 3J–3L). However, it decreased both eEPSC and eIPSC (Fig 3G–3I and 3M–3O). These findings diverged from the observations of Speese and colleagues, who noted no alterations in excitatory evoked release [42], but were consistent with Gracheva and colleagues, who reported a reduction in eEPSCs [47]. The relative decreases in eEPSC amplitude ($\Delta = 2.15$, Fig 3H) and charge transfer ($\Delta = 3.95$, Fig 3I) were greater than those of eIPSC ($\Delta = 1.36$, Fig 3N; $\Delta = 1.54$, Fig 3O) when $I_{WT}/I_{unc-31}$ was analyzed. Therefore, *unc-31(e928)* tilted the E/I balance toward a stronger inhibitory side (Fig 3P), potentially explaining the phenotype of *unc-31 (e928)* mutants, which exhibited a rigid and straight posture indicative of stronger muscle relaxation [41]. Furthermore, the defect in excitatory evoked release in the *unc-31(e928)* mutant could be rescued by expressing UNC-31 cDNA in cholinergic motor neurons (S2 Fig), suggesting that the UNC-31 regulation is largely cell autonomous. Thus, similar to that of CPX-1, the loss of UNC-31 induced different phenotypes in excitatory and inhibitory synapses, with both proteins predominantly regulating excitatory synapses but having a lesser impact on inhibitory synapses.

## Regulation of synaptic vesicle exocytosis by UNC-31 is independent of peptidergic signaling

The alteration in synaptic transmission observed by the deletion of CAPS/UNC-31 has been considered secondary to the dysregulation of DCV release [32,42]. However, the differential reduction in eEPSCs and eIPSCs without affecting spontaneous release of *unc-31(e928)* in our work suggests the direct involvement of UNC-31 in synaptic vesicle exocytosis. To test this hypothesis, we examined whether the phenotypes of *unc-31* null mutants would be occluded by introducing a mutation in EGL-3. EGL-3 is a homolog of mammalian type 2 kex2/subtilisin-like proprotein convertase (KPC-2) and is required for the processing and subsequent maturation of most *C. elegans* neuropeptides [65]. The *egl-3(tm1377)*-null mutant did not show changes in spontaneous release in either excitatory or inhibitory NMJs (Fig 3D–3F and 3J–3L). Importantly, in contrast to the *unc-31* mutation, the *egl-3(tm1377)* mutation did not decrease evoked release from either excitatory or inhibitory synapses (Fig 3G–3I and 3M–3O). These findings suggest that the synaptic transmission defects observed in *unc-31* mutants are at least partially independent of EGL-3-processed peptidergic signaling. Indeed, the *unc-31 (e928); egl-3(tm1377)* double mutant showed no additional changes in eEPSCs or eIPSCs compared with the *unc-31(e928)* single mutant (Fig 3G–3I and 3M–3O).

We further tested the effects of other key enzymes that process peptides, such as KPC-1, AEX-5, and BLI-4, on synaptic vesicle release. KPC-1 is homologous to furin proprotein convertase and is known to process insulin-like peptides [66] and regulate dendritic branching in *C. elegans* [67,68]. *aex-5* encodes a proprotein convertase involved in the secretion of multiple neuropeptides [69–71], whereas *bli-4* encodes kex2/subtilisin-like endoproteases in the proprotein convertase family [72]. However, none of these proprotein convertase family mutants caused spontaneous or evoked release (S3 Fig). Therefore, disruption of neuropeptide synthesis does not negatively affect synaptic transmission, suggesting that the modulation of evoked release by UNC-31 is independent of and extends beyond its involvement in DCV regulation.

### *unc-31* suppressed the enhanced spontaneous release of *cpx-1* mutants at excitatory synapses

Both UNC-31 and CPX-1 are known to differentially regulate synaptic transmission at excitatory synapses. Understanding their interplay could reveal cooperative or antagonistic mechanisms for balancing synaptic vesicle exocytosis. We generated the *cpx-1; unc-31* double mutant and found that it exhibited a striking decrease in body curvature and motility (**S5 Movie**), which was worse than those of the *cpx-1(ok1552)* single mutants (**S2 Movie**). Specifically, the increased body curvature caused by the *cpx-1* mutation was reduced to less than that of the wild-type worms after the *unc-31* mutation was introduced (**Fig 4A–4C**). The double mutants exhibited a posture more like that of *unc-31(e928)* mutants. Additionally, the thrashing frequency in M9 liquid and the free-moving speed on NGM plates further decreased and were even more severe than those in each single mutant (**S4 Fig**). This strong suppression of the *cpx-1* behavioral phenotype by the *unc-31* mutation implies a potential functional interplay between these proteins in synaptic transmission.

We then analyzed excitatory and inhibitory synaptic transmission in the double mutants. Consistent with their motility activity, both eEPSC (**Fig 4D, 4F, and 4G**) and eIPSC (**S5A–S5C**) activities were further decreased in *cpx-1(ok1552); unc-31(e928)* double mutants compared with each single mutant, demonstrating an additive or synergistic effect of *unc-31* and *cpx-1* mutations on evoked release. Similar to the single mutants, the *cpx-1(ok1552); unc-31(e928)* double mutant decreased evoked release in the excitatory synapses more strongly than in the inhibitory synapses (**Fig 4J**). Therefore, these 2 proteins appear to regulate evoked release through distinct pathways in both types of synapses.

Unexpectedly, the enhanced spontaneous release in the excitatory synapses observed in *cpx-1(ok1552)* was abolished in the *cpx-1(ok1552); unc-31(e928)* double mutant (**Fig 4E, 4H, and 4I**). In contrast, mIPSCs were unaffected in the double mutant (**S5D–S5F**), which is consistent with the lack of effects of *unc-31(e928)* and *cpx-1(ok1552)* on mIPSCs. Hence, the *unc-31* null mutation restored the E/I imbalance of the *cpx-1* mutant during spontaneous release. However, regarding evoked release, the more severe impairment of excitatory synaptic transmission in the double mutant shifted the balance toward inhibition, leading to excessive muscle relaxation and reduced body curvature (**Fig 4J**).

Next, we examined whether the *unc-31* null mutation also suppressed the enhanced spontaneous release of the CPX-1 clamping-specific mutant *cpx-1(Δ12)*. Consistent with our observations in the *cpx-1(ok1552)* null mutant, the increase in mPSC in the *cpx-1(Δ12)* mutant was abolished in the *cpx-1(Δ12); unc-31(e928)* double mutant (**Fig 4E, 4H and 4I**). Additionally, the *cpx-1(Δ12)* clamping mutation further decreased both eEPSC (**Fig 4D, 4F and 4G**) and eIPSC (**S5A–S5C Fig**) in the *unc-31(e928)*-null background. Similarly, *cpx-1(Δ12); unc-31* double mutant exhibited a decrease in body curvature and motility (**Fig 4A–4C and S6 Movie**) compare to those of the *cpx-1(Δ12)* single mutants (**S3 Movie**). This result confirms that UNC-31 was indispensable for the enhanced spontaneous release induced by the removal of the CPX-1 clamping function.

### The suppression of *cpx-1* by *unc-31* relies on the presence of external Ca$^{2+}$

The spontaneous release of synaptic vesicles (SVs) requires external Ca$^{2+}$. We sought to examine the necessity of external Ca$^{2+}$ on the enhanced mPSC in the *cpx-1* mutant and the role of UNC-31. At 5 mM external Ca$^{2+}$, the mPSC frequency in the *cpx-1* mutant was approximately twice that in the wild-type animals (**Fig 5A and 5B**). The removal of external Ca$^{2+}$ (0 mM) reduced the mPSC frequency in both *cpx-1* mutant and wild-type animals. However, the mPSC frequency and amplitude of the *cpx-1* mutant were still significantly greater than those

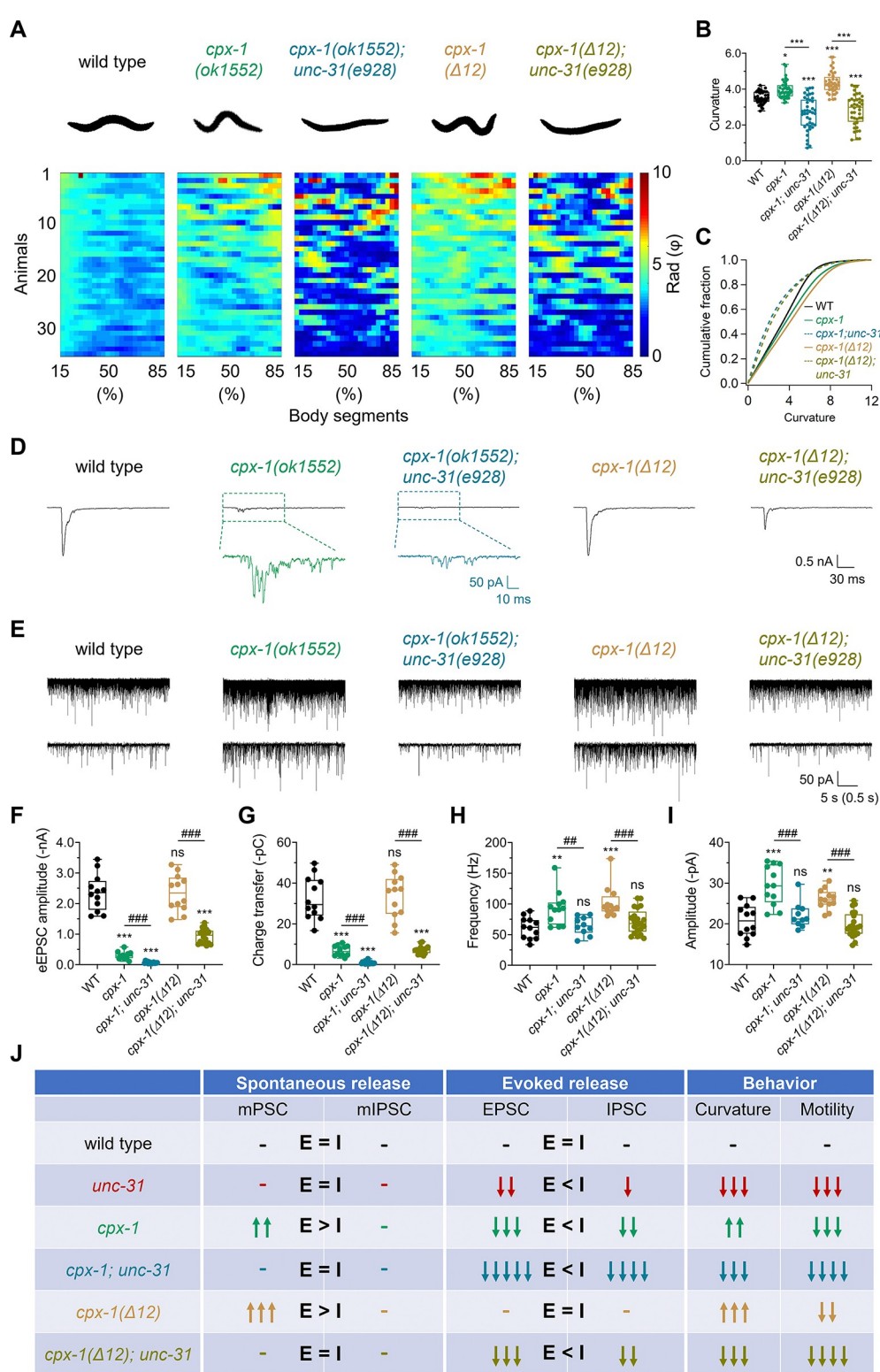

**Fig 4. UNC-31 regulates enhanced spontaneous release in the excitatory synapses of *cpx-1* mutants.** (**A**) Representative body bending states (top) and color map of curvature (bottom) in wild-type, *cpx-1(ok1552)*, *cpx-1 (ok1552); unc-31(e928)*, *cpx-1(Δ12)*, and *cpx-1(Δ12); unc-31(e928)* worms. (**B**, **C**) Quantification (**B**) and distribution (**C**) of body curvature under the indicated strains. $n \geq 39$ animals. One-way ANOVA was performed ($F_{(4, 195)} = 47.65$, $P < 0.0001$ for **B**). (**D**) Representative traces of eEPSCs recorded from the indicated genotypes. The lower traces of *cpx-*

*1(ok1552)* (green) and *cpx-1(ok1552); unc-31(e928)* (blue) show a zoomed-in region from the upper traces. (**E**) Representative traces of mPSCs recorded from the mentioned genotypes, displayed at different timescales. (**F, G**) Quantification of eEPSC amplitude (**F**) and charge transfer (**G**) at each strain. $n \geq 12$ animals. One-way ANOVA was performed ($F_{(4, 64)} = 108.8$, $P < 0.0001$ for **E**; $F_{(4, 64)} = 80.13$, $P < 0.0001$ for **G**). (**H, I**) Quantification of the mPSC frequency (**H**) and amplitude (**I**) in each genotype. $n \geq 10$ animals. One-way ANOVA was performed ($F_{(4, 62)} = 8.581$, $P < 0.0001$ for **H**; $F_{(4, 62)} = 18.25$, $P < 0.0001$ for **I**). (**J**) Summary graph showing the differential regulation of excitatory and inhibitory synaptic vesicle release in the indicated genotypes. The data are presented as box-and-whisker plots, with the median (central line), 25th–75th percentile (bounds of the box), and 5th–95th percentile (whiskers) indicated. Student's *t* test was performed for comparisons of 2 groups, ## $P < 0.01$; ### $P < 0.001$. One-way ANOVA was used for comparisons of multiple groups, followed by Tukey's range test, * $P < 0.05$; ** $P < 0.01$; *** $P < 0.001$; ns, not significant. The error bars represent the SEM. $N = 3$ independent replicates. All the raw data associated with this figure are available in S1 Data.

of the wild-type animals (**Fig 5A–5C**). Compared with 5 mM external $Ca^{2+}$, pre-incubation with 1 mM BAPTA-AM in $Ca^{2+}$-free medium lowered the mPSC frequency approximately threefold in both wild-type and *cpx-1* animals (**Fig 5A and 5B**). However, the increased mPSC amplitude in the *cpx-1* mutant remained unchanged, indicating independence from external $Ca^{2+}$ (**Fig 5C**). Thus, consistent with prior studies [31], the enhanced mPSCs observed in the *cpx-1* mutant are $Ca^{2+}$ dependent.

Given that UNC-31 plays a crucial role in maintaining the enhanced spontaneous release observed in *cpx-1* mutants, we hypothesized that UNC-31 functions as a secondary $Ca^{2+}$ sensor in addition to synaptotagmin-1, as previously suggested from complexin knockdown experiments [31]. To test this hypothesis, we investigated the $Ca^{2+}$ dependency of the regulatory effect of UNC-31 on *cpx-1* mutants. At external $Ca^{2+}$ concentrations ranging from 0.5 mM to 1.5 mM, the wild-type and *unc-31* single mutants presented comparable mPSC patterns. In contrast, the increased mPSC frequency and amplitude observed in the *cpx-1* mutant were abolished in the *cpx-1; unc-31* double mutant at the same external $Ca^{2+}$ concentration, which was consistent with observations at 5 mM external $Ca^{2+}$. Unexpectedly, at zero external $Ca^{2+}$, the *cpx-1; unc-31* double mutant displayed a similar level of enhanced mPSC frequency and amplitude as the *cpx-1* mutant did (**Fig 5D–5F**). These results imply that the suppression of the *cpx-1* mutant by the *unc-31* mutation depends on the presence of external $Ca^{2+}$. Alternatively, CAPS/UNC-31 could function as a $Ca^{2+}$-sensor responsible for spontaneous release, a role that is hindered by the presence of complexin/CPX-1.

## CAPS/UNC-31 and complexin/CPX-1 do not bind directly

Next, to investigate whether the co-regulation of spontaneous release by CAPS/UNC-31 and complexin/CPX-1 is due to a direct interaction, we conducted a series of experiments. Previous studies have shown that CAPS can bind to the SNARE proteins like syntaxin-1, SNAP-25, and VAMP-2 through its Munc13 homology domain-1 (MHD1), which is essential for CAPS function in dense-core vesicle exocytosis [73,74]. Additionally, recent findings indicate that CAPS can directly interact with Munc13-1 via the DAMH domain [44]. However, no interaction between CAPS and complexin has been reported.

To test for a potential direct interaction, we generated a full-length CAPS and performed glutathione S-transferase (GST) pull-down experiments using GST-complexin (see **Materials and methods**). No CAPS was pulled down in the presence or absence of $Ca^{2+}$ (**Fig 5G**), suggesting that no direct binding between CAPS and complexin.

We further speculated that any interaction might be dependent on other components of the exocytosis machinery, such as the SNARE complex. Additional GST pull-down experiments using GST-SNARE (either assembled or disassembled) showed that while both complexin and trace amounts of full-length CAPS could bind to the SNARE complex, there was no strong

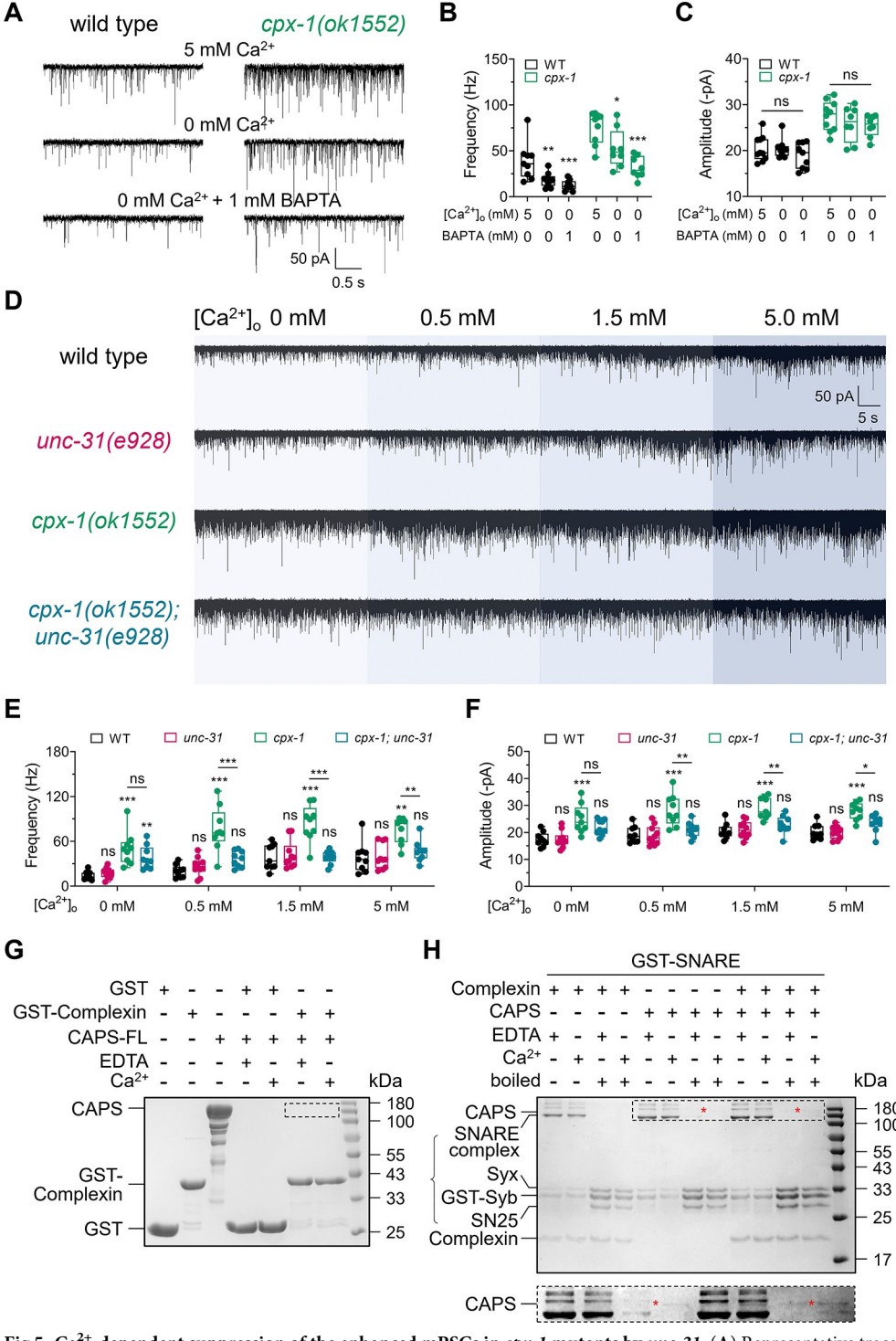

**Fig 5. Ca²⁺-dependent suppression of the enhanced mPSCs in *cpx-1* mutants by *unc-31*.** (**A**) Representative traces of mPSCs recorded from wild-type and *cpx-1(ok1552)* worms in a bath solution containing 5 mM $[Ca^{2+}]_o$, 0 mM $[Ca^{2+}]_o$, or 0 mM $[Ca^{2+}]_o$ with 1 mM BAPTA-AM. (**B, C**) Quantification of the mPSC frequency (**B**) and amplitude (**C**) in the WT and *cpx-1(ok1552)* strains in various bath solutions. An evident Ca²⁺ dependence of enhanced mPSCs in *cpx-1* mutants was observed. $n \geq 8$ animals. One-way ANOVA was performed ($F_{(2, 24)} = 10.26$, $P = 0.0006$ for WT, and $F_{(2, 22)} = 13.43$, $P = 0.0002$ for *cpx-1* in **B**; $F_{(2, 24)} = 0.9251$, $P = 0.4102$ for WT, and $F_{(2, 22)} = 1.568$, $P = 0.2309$ for *cpx-1* in **C**). (**D**) Representative traces of mPSCs recorded from WT, *unc-31(e928)*, *cpx-1(ok1552)*, and *cpx-1(ok1552); unc-31 (e928)* worms in different Ca²⁺ bath solutions. (**E, F**) Quantification of the mPSC frequency (**E**) and amplitude (**F**) in response to the indicated strains. The suppression of the enhanced spontaneous release in *cpx-1* mutants by *unc-31* was

apparent only in the presence of $[Ca^{2+}]_o$. $n \geq 9$ animals. One-way ANOVA was performed ($F_{(3, 33)} = 13.55$, $P < 0.0001$ for 0 mM; $F_{(3, 33)} = 19.09$, $P < 0.0001$ for 0.5 mM; $F_{(3, 33)} = 13.95$, $P < 0.0001$ for 1.5 mM; $F_{(3, 33)} = 8.934$, $P = 0.0002$ for 5 mM in **E**. $F_{(3, 33)} = 9.683$, $P < 0.0001$ for 0 mM; $F_{(3, 33)} = 11.14$, $P < 0.0001$ for 0.5 mM; $F_{(3, 33)} = 12.94$, $P < 0.0001$ for 1.5 mM; $F_{(3, 33)} = 15.23$, $P < 0.0001$ for 5 mM in **F**). (**G**) Pull-down experiments indicate no detectable direct binding of full-length CAPS (CAPS-FL) to GST-Complexin, regardless of the presence or absence of $Ca^{2+}$. The dashed box highlights the absence of CAPS-FL in the pull-down. (**H**) Top: Binding analysis of complexin and CAPS-FL to GST-SNARE complexes, whether assembled or disassembled by boiling, in both the presence and absence of $Ca^{2+}$. Bottom: A magnified view of the region outlined by the dashed box in the top panel, showing the specific bands corresponding to CAPS, marked by red asterisks. The data are presented as box-and-whisker plots, with the median (central line), 25th–75th percentile (bounds of the box), and 5th–95th percentile (whiskers) indicated. One-way ANOVA was performed, $^*$ $P < 0.05$; $^{**}$ $P < 0.01$; $^{***}$ $P < 0.001$; ns, not significant. The data are presented as the means ± SEMs. $N = 3$ independent replicates. All the raw data associated with this figure are available in S1 Data. Western blot can be found in S1 Raw Images. mPSC, miniature postsynaptic current.

evidence of direct binding between CAPS and complexin (**Fig 5H**). The presence of CAPS did not enhance SNARE binding to complexin, nor did complexin enhance CAPS binding to SNARE, although weak, calcium-dependent binding between CAPS and the SNARE complex was observed (**Fig 5H**).

These results suggest that CAPS and complexin do not directly bind to each other. Their regulation of synaptic vesicle exocytosis is likely mediated through indirect interactions involving other components of the exocytosis machinery.

## Activity-dependent synaptic plasticity was suppressed by *cpx-1* and rescued by *unc-31*

Complexin plays a crucial role in regulating vesicle pool size, including the readily releasable pool, which is essential for short-term synaptic response changes [21]. To further investigate the conserved role of UNC-31's dependence on CPX-1 in regulating synaptic neurotransmission, we utilized optogenetic stimulation protocols to induce activity-dependent synaptic plasticity specifically at excitatory synapses [57,75]. We employed 2 protocols: one involving 10 trains of short (0.1 s) light pulses delivered every 0.5 s (2 Hz) to mimic modest nerve stimulation, and another using 10 trains of longer (1 s) light pulses delivered every 5 s (0.2 Hz) to simulate tonic stimulation [76,77] (**Fig 6A**). These protocols targeted excitatory neuromuscular junctions through the expression of channelrhodopsin-2 (ChR2) in cholinergic motor neurons. We found that in wild-type animals, the modest stimulation induced a transient increase in miniature postsynaptic current (mPSC) frequency, which peaked at 10 s post-stimulation and rapidly returned to baseline (**Fig 6A and 6B**). By contrast, tonic stimulation resulted in a sustained increase in spontaneous release, as shown by a significantly elevated mPSC frequency lasting over 40 s (**Fig 6A and 6B**). These results indicate that spontaneous mPSCs demonstrates stimulus-dependent synaptic plasticity.

We then tested whether synaptic plasticity was affected in *cpx-1* and *unc-31* mutants. Given the sustained changes observed with tonic stimulation, we focused on analyzing mPSC frequency at different time intervals following this protocol. Unexpectedly, in the *cpx-1(ok1552)* mutant, tonic stimulation not only failed to increase mPSC frequency but instead caused a significant reduction, which persisted for over 40 s post-stimulation (**Fig 6C and 6D**). The mechanism behind this inhibition is unclear, but it could be due to a substantial depletion of synaptic vesicles, which cannot be replenished promptly in the presynaptic readily releasable pool following repeated tonic stimuli [22]. More interesting, the mPSC inhibition seen in *cpx-1(ok1552)* mutants was suppressed in the presence of *unc-31* mutation. In the *cpx-1(ok1552); unc-31(e928)* double mutant, the response was neither enhanced nor inhibited, resembling the *unc-31* mutant alone (**Fig 6C and 6D**) [57]. These results show that the activity-dependent

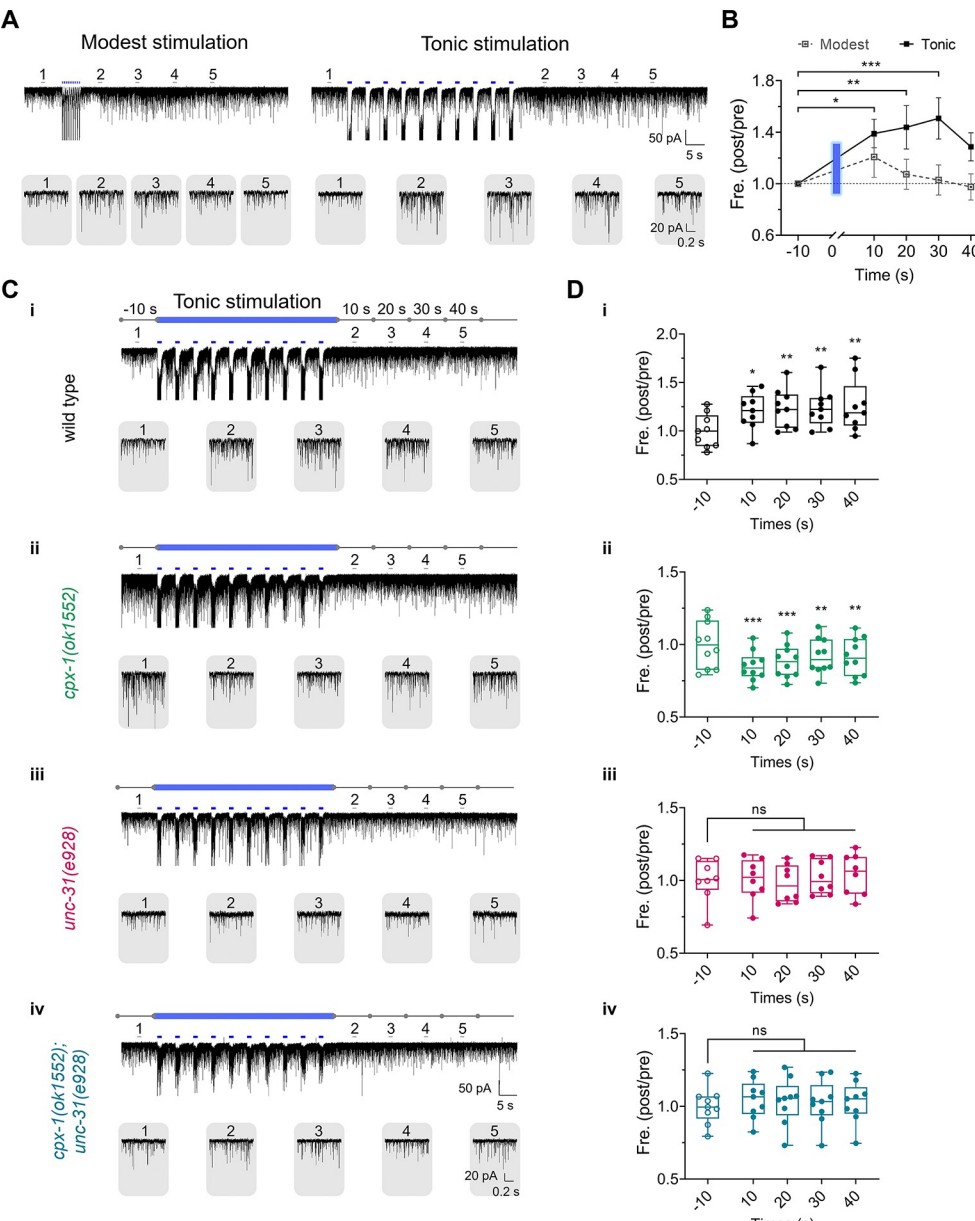

**Fig 6. The UNC-31 requirement for stimulus-dependent spontaneous release increased.** (**A**) Sample traces of persistent mPSCs before and after stimulation with repeated blue light illumination (blue bars) from wild-type worms. Modest stimulation: 10 × 0.1 s pulses at 2 Hz. Tonic stimulation: 10 × 1 s pulses at 0.2 Hz. Zoomed-in views of individual minis traces are presented in the gray rounded rectangle, extracted from the indicated regions of the upper panel (lines 1–5). (**B**) Ratio between mPSC frequency post- and pre-blue light illumination in (**A**). The blue bar depicts the duration of 10 consecutive illuminations. $n \geq 8$ animals. (**C**) Sample traces of persistent mPSCs before and after tonic stimulation from wild-type (i), *cpx-1(ok1552)* (ii), *unc-31(e928)* (iii), and *cpx-1(ok1552); unc-31(e928)* (iv) worms. (**D**) Ratio between mPSC frequency post- and pre-blue light illumination in (**C**). $n \geq 8$ animals. Two-way ANOVA was performed, * $P < 0.05$; ** $P < 0.01$; *** $P < 0.001$; ns, not significant. The error bars represent the SEMs. $N = 3$ independent replicates. All the raw data associated with this figure are available in S1 Data. mPSC, miniature postsynaptic current.

synaptic inhibition observed in *cpx-1* mutants also relies on UNC-31, further supporting the conserved role of CAPS/UNC-31 in modulating complexin/CPX-1 regulation of synaptic neurotransmission.

In summary, both complexin/CPX-1 and CAPS/UNC-31 showed a preferential regulation of excitatory synapses. The dependence of complexin/CPX-1 regulation on CAPS/UNC-31 extends beyond spontaneous release to also include stimulus-dependent synaptic plasticity.

## Discussion

Our understanding of SNARE-dependent synaptic vesicle release has primarily come from studies on excitatory synapses, leading to the assumption that the regulatory mechanisms in these synapses are directly applicable to inhibitory neurons. However, our findings reveal a significant differential regulation of synaptic release by CPX-1 and UNC-31 between excitatory and inhibitory synapses. We demonstrated that both CPX-1 and UNC-31 function predominantly in excitatory synapses. Notably, the *unc-31* null mutant severely impairs evoked release in excitatory synapses, independent of its role in dense-core vesicle regulation. Additionally, the increased spontaneous release observed in the *cpx-1* null mutant was specific to excitatory synapses and required UNC-31. Given the established functional conservation of complexin/CPX-1 and CAPS/UNC-31, our findings are pivotal for understanding the mechanisms governing synaptic secretion, E/I balance, and behavioral outputs in different species.

Several factors may explain the biased regulation of CPX-1 and UNC-31 in excitatory versus inhibitory synapses. First, the size of the readily releasable pool (RRP) of synaptic vesicles differs between these synapse types. For instance, in striatal inhibitory GABAergic neurons, the RRP size is 3 times larger than that in excitatory hippocampal glutamatergic neurons, as demonstrated by hypertonic sucrose experiments [78,79]. These quantitative differences influence vesicle dynamics, release, and recycling, suggesting that the clamping role of CPX-1 may have distinct impacts depending on the synapse type. Second, voltage-gated calcium channels (VGCCs) exhibit different regulatory functions in spontaneous release. Inhibitory synapses rely on redundant roles of VGCCs, while in excitatory synapses, UNC-2/CaV2 and EGL-19/CaV1 channels independently regulate spontaneous release. In addition, $Ca^{2+}$-independent spontaneous release is more pronounced in inhibitory synapses, indicating that excitatory synapses depend more on VGCCs for vesicle release [80]. This greater reliance positions CPX-1 as a crucial calcium sensor in excitatory synapses, amplifying its regulatory effects. Finally, previous research suggests that the composition and regulatory interactions of the SNARE complex vary between excitatory and inhibitory synapses [81–83]. Complexin-1/CPX-1 stabilizes the SNARE complex but may interact differently depending on the specific SNARE isoforms or auxiliary proteins present. For example, excitatory synapses predominantly utilize syntaxin-1A, whereas inhibitory synapses may express higher levels of syntaxin-1B [80,84,85]. Furthermore, the distinct $Ca^{2+}$ sensors involved in spontaneous release regulation between these synapses [86] could result in different affinities and regulatory effects of CPX-1 and UNC-31. These insights underline the necessity for further investigation into exocytosis mechanisms in inhibitory synapses.

Complexin can facilitate evoked release and clamp spontaneous asynchronous release [23,25]. In this study, we showed more new characterizations of how complexin regulates exocytosis. We found out that CPX-1 functions depend on synapses. In the *cpx-1* null mutant, evoked release is decreased in both excitatory and inhibitory synapses, whereas spontaneous release is increased only in excitatory synapses. Our findings are consistent with those of previous studies on complexin in mice and *Drosophila*, which revealed that the most consistent phenotype of complexin mutants is reduced evoked release rather than enhanced spontaneous release in excitatory transmission [15,21,25]. With respect to GABAergic transmission, complexin knockout in mammalian striatal GABAergic neuronal cultures resulted in a decrease in evoked and spontaneous IPSCs [15]. Our results aligned with the decreased evoked release but

were inconsistent with the effects observed in spontaneous release. These inconsistencies come partly from differences across species, where invertebrate CTDs have stronger inhibitory effects than mammalian complexin do [51,87]. To this end, species variation and the structural basis of the differences in the functions of complexin in controlling the release of inhibitory synaptic vesicles need to be clarified. As locomotion depends on the E/I balance at the NMJ [88], the tilted balance toward increased inhibition explains the impaired motility in the *cpx-1* mutant. A reduction in complex protein levels is found in patients with schizophrenia, major depressive disorder, and neurodegenerative diseases such as Huntington's disease and Alzheimer's disease [89]. Hence, we propose that the loss of complexin alters the E/I balance in neural circuits as a mechanism of brain disorders.

The clamping function of complexin/CPX-1 is considered a critical factor in preventing the depletion of primed vesicles that allow synchronized evoked release [17,90]. However, our clamping-specific mutant, *cpx-1(Δ12)*, showed normal evoked release despite the dramatically increased spontaneous release in the excitatory synapse and a modest increase in the inhibitory synapse (**Fig 2**). A similar clamping-specific mutation has been reported in *Drosophila*, although its ability to regulate spontaneous effects is spliced isoform dependent [25,91]. We found that the clamping of spontaneous release is not a prerequisite to support evoked release. Works in both mammalian and *Drosophila* excitatory and inhibitory synapses suggest that spontaneous and evoked release can be segregated, with differential pools of synaptic vesicles for each mode of release at the active zone [92,93]. Hence, CPX-1 might also participate in the regulation of vesicle release in both pools. Previous biochemical work has shown that synaptotagmin is the regulatory protein that removes the clamping complexin to trigger $Ca^{2+}$-dependent evoked synchronized release [11,12,94]. To our surprise, genetic evidence to support this well-known hypothesis is lacking and contradictory. A double mutant in *Drosophila* with the loss of both complexin and synaptotagmin showed an additive effect in impairing spontaneous and evoked release, indicating partly independent regulatory functions [21]. However, in mice, the synaptotagmin $Ca^{2+}$-binding C2B domain can clamp synaptic vesicles independent of complexin [95]. More work is necessary to clarify the genetic and functional interaction between synaptotagmin and complexin.

Notably, the mPSC amplitude increase in both the *cpx-1* null mutant and the clamping-specific mutant might suggest a more pronounced effect on the size or content of individual synaptic vesicles, a detail that might have been less emphasized or observed in previous studies [43,96–98]. The increased mPSC amplitudes might reflect changes in the properties or the state of the vesicle pools. For example, the absence of complexin could lead to the preferential release of larger vesicles or vesicles with a higher neurotransmitter load, or it may lead to the release of multiple vesicles, thereby increasing the amplitude of individual mPSCs [99,100]. Alternatively, the deletion of CPX-1 may have caused alterations in postsynaptic receptors, including activity-dependent changes or developmentally relevant regulation [101–104], resulting in more excitatory postsynaptic receptors clustered at the postsynaptic membrane. The mechanism by which complexin regulates secretion kinetics needs to be further investigated.

UNC-31 is well known for its role in the DCV regulation of neuropeptide release, but its role in synaptic vesicle regulation is controversial [32,34,39,42,105]. We found that the *unc-31* null mutation decreased evoked exocytosis without affecting spontaneous release, and this decrease was more evident in excitatory synapses. Previously, the disrupted synaptic transmission of *unc-31* mutants was considered secondary to defects in DCV secretion [47]. However, we found that none of the mutants that play important roles in DCV maturation or release mimicked or occluded the phenotype of *unc-31*. Therefore, the role of UNC-31 in directly regulating synaptic vesicle release warrants reconsideration. A recent study established a functional propagation atlas in *C. elegans* and proposed that UNC-31 is critical for extrasynaptic

transmission via DCVs only [7]. While the authors demonstrated that certain functional connections were lost in the *unc-31* null mutant, we challenged the notion and assumption that UNC-31 is necessary only for DCV release. Independent of DCVs, the loss of neural signal transmission observed by Randi and colleagues could be explained by a strong decrease in evoked excitatory transmission via indirect pathways involving multiple neurons. Thus, while extrasynaptic signaling is fundamental in driving neuronal activities, careful precautions on the contribution of DCV pathways need to be taken when UNC-31 is assumed to have no direct impact on synaptic vesicle exocytosis in *C. elegans*. The *unc-31* mutant is immobile and appears in a relaxed posture [41]. On the basis of our findings, we propose that tilted balance toward stronger inhibition drives this impaired motor behavior.

Interestingly, although the *unc-31* null mutation did not affect spontaneous release, it blocked the enhanced spontaneous release caused by *cpx-1* mutants as well as by repetitive depolarization stimuli. UNC-31 might play a critical role in short-term plasticity and synaptic depression. CAPS-1 deletion in the thalamocortical synapse caused earlier synaptic depression under repetitive stimuli [106]. This finding is consistent with our observation that under tonic stimulation, *unc-31* null was unable to increase spontaneous synaptic vesicle release. Thus, it is tempting to postulate that UNC-31 is important not only for evoked synaptic release but also for the maintenance of enhanced spontaneous release under synaptic plasticity and in synapses with a high release probability.

Another important mechanistic question is how UNC-31 regulates the increased spontaneous release observed in *cpx-1* mutant. Acting as a positive regulator of exocytosis, complexin selectively binds to the neuronal SNARE complex. It also acts as a negative regulator by anchoring vesicles to prevent fusion. Its N-terminal alpha-helix domain integrates into the SNARE complex helix bundle, inhibiting the complete zippering of the assembly [21,107]. Complexin knockdown has been associated with synaptotagmin-1 and -10-dependent exocytosis [108]. Unlike synaptotagmin, our experiments show that UNC-31 is selectively required for maintaining the enhanced spontaneous release in *cpx-1* mutants. Moreover, UNC-31 does not form direct protein complexes with complexin, either with or without the SNARE complex, suggesting UNC-31 facilitates this enhanced spontaneous release through an indirect mechanism. Revealing this CAPS/UNC-31 dependence deepens our understanding of the dual regulatory role of complexin/CPX-1 in synaptic vesicle exocytosis.

Together, by conducting precise electrophysiological analyses on both excitatory and inhibitory synapses, we uncovered surprising E/I imbalances in previously studied synaptic exocytosis mutants, as well as in our new clamping-specific knockin mutant. These findings highlight the importance of examining inhibitory synaptic release alongside excitatory synapses, as their regulatory mechanisms may differ significantly. As we showed from 2 conserved exocytosis regulators, complexin/CPX-1 and CAPS/UNC-31, each of them has a different ability to control evoked and spontaneous synaptic vesicle exocytosis, ultimately shaping the synaptic E/I balance. Multi-technique analysis further supports the conserved role of CAPS/UNC-31 in modulating complexin/CPX-1 regulation of synaptic neurotransmission, providing a more comprehensive understanding of mutant behavior. With increasing knowledge of the regulation of excitatory and inhibitory synapses, we will identify better therapeutic targets for modulating E/I balance in neural circuits linked to neurological and psychiatric disorders.

## Materials and methods

### Strains

The complete lists of constructs, primers, transgenic lines, and strains generated or acquired for this study are provided in **S1**–**S3** **Tables**. PHX3665 *cpx-1(syb3665) I* and PHX3584 *cpx-1*

*(syb3584) I* were generated by SunyBiotech via CRISPR/Cas9 sequence knock-in. The generation of transgenic strains that carry extrachromosomal arrays (*gaaEx*) involved the co-injection of plasmid DNA with a marker. All *C. elegans* strains were cultured on standard nematode growth medium (NGM) plates seeded with OP50 and maintained at 22˚C. Unless otherwise noted, the wild-type animal refers to the Bristol N2 strain. Only hermaphrodite worms were used for the experiments.

## Molecular biology

All expression plasmids were constructed with a three-fragment multisite gateway system (Invitrogen, Thermo Fisher Scientific, Waltham, Massachusetts, United States of America). Three entry clones, A, B, and C, were recombined into the pDEST R4-R3 Vector II destination vectors via standard attL-attR (LR) recombination reactions to generate expression clones.

The entry clone A slot1 was generated via the In-Fusion method via the ClonExpress One Step Cloning Kit (Vazyme, Nanjing). The acetylcholine neuron promoter *unc-17* (3262 bp) fragment was amplified from N2 *C. elegans* genomic DNA to replace the *rab-3* fragment in the standard BP reaction-generated entry clone A.

All entry clones B slot2 and C slot3 were generated via BP recombination reactions. The full-length cDNA (4126 bp) encoding UNC-31 was combined into the pDONR221 donor vector as slot2. The fluorescent protein *sl2d*-Wcherry was combined with the pDONR-P2R-P3 donor vector to generate slot3.

## Aldicarb assays

Aldicarb sensitivity was assessed in synchronously growing adult worms placed on non-seeded 30 mm NGM plates containing 1 mM aldicarb. Over a 4- or 24-h period, worms were monitored for paralysis at 15- or 30-min intervals. Worms were considered paralyzed when there was no movement or pharyngeal pumping in response to 3 taps to the head and tail with a platinum wire. Once paralyzed, the worms were removed from the plate. Six sets of 15 to 20 worms were examined for each strain.

## Thrashing assays

The motility of each strain was determined by counting the thrashing rate of *C. elegans* in liquid M9 medium. Briefly, worms were bleached to release their eggs and were then synchronously grown to young adulthood. Young adult worms were placed in a 60 μl drop of M9 buffer on a 30 mm Petri dish cover. After a 2-min recovery period, the worms were video recorded for 2 min via an OMAX A3580U camera on a dissecting microscope with OMAX ToupView software. The number of thrashes per minute was manually counted and averaged within each strain. A thrash was defined as a complete bend in the opposite direction at the midpoint of the body. In each experiment, at least 40 worms were measured for each strain.

## Velocity and body curvature assays

Velocity and body curvature assays were performed with L4 stage animals. Plates for behavioral assays were prepared from 60 mm NGM plates seeded with a thin layer of OP50. Before the experiment, the OP50 lawn was spread evenly across the plate with a sterile bent glass rod. A single worm was then gently transferred to an assay plate with eyebrows. One minute after acclimatization, a 20-s movie of spontaneous locomotion was recorded on a modified stereomicroscope (Axio Zoom V16, Zeiss) with a digital camera (acA2500-60um, Basler). All images were captured with a 10× objective at 10 Hz.

Behavioral parameter analysis after imaging was performed via an in-house written MATLAB (MathWorks, Inc., Natick, Massachusetts, USA) script. The central line was used for tracking. The midpoint was used to calculate the velocity between each frame. Images for curvature analysis from each animal were divided into 32 body segments. The curvatures at each point along the worm centerline $k(s)$ can be calculated with the coordinate of each point $(x(s), y(s))$ via the formula $k(s) = \frac{x'y''-x''y'}{(x'^2+y'^2)^{(3/2)}}$, where s is the normalized location along the centerline (head = 0, tail = 1), and the unit of $k$ is pixel^(-1). Then, $k$ is normalized with the length of the worm body $L$, resulting in the dimensionless $\tilde{k}\sim(s)$ via the formula $\tilde{k}(s) = k(s) \times L$. To exclude errors due to uncoordinated head and tail oscillations, the curvature was calculated for 15% to 85% of the body segments. At least 40 worms of each strain were imaged and analyzed in each set of experiments.

### In situ electrophysiology

Dissection and recording were carried out via protocols and solutions described in previous studies [58,109]. Briefly, 1- or 2-day-old hermaphrodite adults were glued (Histoacryl Blue, Braun) to a Sylgard (Dow Corning, USA)-coated cover glass covered with bath solution. Under a DIC microscope, semi-fixed worms were dissected dorsally with a glass pipette, the cuticle flap was flipped and gently glued (WORMGLU, GluStitch Inc.) to the opposite side, and the intact ventral body muscles were exposed after the viscera were cleaned. Anterior body wall muscle cells were patched with 4–6 MΩ resistant borosilicate pipettes (World Precision Instruments, USA), which were pulled with a micropipette puller P-1000 (Sutter). Membrane currents were recorded in the whole-cell configuration via PULSE software with an EPC-9 patch clamp amplifier (HEKA, Germany). The data were digitized at 10 kHz and filtered at 2.6 kHz. The pipette mixture contained the following (in mM): K-gluconate 115; KCl 25; $CaCl_2$ 0.1; $MgCl_2$ 5; BAPTA 1; HEPES 10; $Na_2ATP$ 5; $Na_2GTP$ 0.5; cAMP 0.5; cGMP 0.5, pH 7.2 with KOH, ~320 mOsm. cAMP and cGMP were included to maintain the activity and longevity of the preparation. The bath solution consisted of the following (in mM): NaCl 150; KCl 5; $CaCl_2$ 5; $MgCl_2$ 1; glucose 10; sucrose 5; and HEPES 15, pH 7.3 with NaOH, ~330 mOsm. Different extracellular $Ca^{2+}$ bath solutions were prepared according to the bath solution with different $CaCl_2$ concentrations (0, 0.5, 1.5, or 5 mM). Zero $Ca^{2+}$ with BAPTA solution consists of an additional 1 mM BAPTA in the 0 $Ca^{2+}$ bath solution to further chelate and reduce the amount of extracellular free $Ca^{2+}$ [31]. The muscle cells were held at −60 mV when the mPSCs or eEPSCs were recorded. To isolate mIPSCs or eIPSCs, recordings were performed with a holding potential of −10 mV, with 0.5 mM D-tubocurarine (d-TBC) included in the bath solution to block all acetylcholine receptors [58,59]. Chemicals were obtained from Sigma unless stated otherwise. The experiments were performed at room temperature (20 to 22°C).

### Optogenetics

NGM plates were seeded with OP50 bacteria containing 500 μm all-trans retinal (Sigma). The seeded retinal plates were kept in darkness at 4°C. L4 worms were transferred from regular plates to retinal plates and then grown for 24 h in the dark before electrophysiological recording. Light stimulation of *zxIs6* or *zxIs3* was performed with an LED light source at a wavelength of 460 ± 5 nm (13.75 mW/cm²) to evoke postsynaptic currents [57,110]. The light pulse was triggered by PULSE software with a duration of 10 ms unless otherwise stated. In **Fig 6**, 10 light stimuli of 0.1 s are applied at a frequency of 2 Hz (modest), or 10 light stimuli of 1 s are applied at a frequency of 0.2 Hz (tonic) to evoke a series of eEPSCs.

## Plasmid, protein purification, and GST pull-down experiments

Full-length CAPS-1 was expressed and purified as previously described [44]. The plasmid pFastBac-CAPS-1 for eukaryotic expression was generated by sub-cloning CAPS-1 from a pcDNA3.1-CAPS-1 plasmid into the pFastBacHtB vector (Invitrogen), which contains a polyhedron promoter (for high-level expression of recombinant protein in insect cells) before the start codon and encodes an N-terminal TEV cleavable 6× His-tag. Full-length CAPS-1 was expressed in Sf9 insect cells as described previously [111]. Sf9 cells were cultured in ESF 921 medium (Expression Systems) at 27°C at 110 rpm. Sf9 insect cells were infected with CAPS-1 baculovirus, harvested approximately 72 h post infection, and resuspended in lysis buffer (50 mM Tris (pH 8.0), 300 mM NaCl, and 10 mM imidazole). The cells were lysed and then centrifuged at 16,000 rpm for 30 min, and the supernatant was incubated with $Ni^{2+}$-NTA agarose (QIAGEN) at 4°C for 2 to 3 h. The beads were washed with lysis buffer supplemented with an additional 10 to 30 mM imidazole. The protein was eluted with 300 mM imidazole and further purified by gel filtration in buffer containing 25 mM HEPES (pH 7.4), 150 mM KCl, and 10% glycerol (v/v).

Full-length rat complexin (residues 1–134), full-length human SNAP-25 (SN25) (residues 1–206), and the cytoplasmic domain of Syb2 (residues 29–96) were inserted into the pGEX-6p-1 vector (GE Healthcare), which contains a PreScission Protease cleavable N-terminal GST tag. All the recombinant GST fusion proteins described above were expressed in the *E. coli* BL21 (DE3) strain cultured in LB media at 37°C to an OD600 of 0.8 and were induced with 0.4 mM IPTG at 25°C for 16 h. The cells were lysed in Tris buffer A [20 mM Tris (pH 8.0) and 1 M NaCl] supplemented with 0.5% Triton X-100, 5 mM EDTA, and 1 mM PMSF and then disrupted via an AH-1500 Nano Homogenize Machine (ATS Engineering, Inc.) at 800 bar 3 times at 4°C. The cell lysates were subsequently centrifuged at 16,000 rpm for 30 min in a JA-25.50 rotor (Beckman Coulter) at 4°C. The supernatants were incubated with 2 ml of glutathione-Sepharose beads (Amersham-Pharmacia Biotech) at 4°C for 3 h. The bound proteins were washed with Tris buffer A. The GST tag was cleaved from the proteins by GST-fused PreScission protease (10 U/mg protein) at 4°C overnight on the beads. For purification of GST-Complexin-1 and GST-Syb2, proteins were eluted with Tris buffer B [20 mM Tris (pH 8.0) and 150 mM NaCl] containing 20 mM L-glutathione (GSH; Amresco). The proteins were further purified by size-exclusion chromatography on a Superdex 75 pg 16/600 column (GE Healthcare) and stored in Tris buffer B.

The cytoplasmic domain of rat Syx1 (residues 1–261) was inserted into the pET28a vector (Novagen). The cells were lysed in Tris buffer A containing 0.5% Triton X-100, 5 mM EDTA, and 1 mM PMSF and then centrifuged at 16,000 rpm for 30 min, after which the supernatant was incubated with $Ni^{2+}$-NTA agarose (QIAGEN) at 4°C for 2 h. The beads were washed with the lysis buffer described above supplemented with an additional 30 mM imidazole. The protein was eluted with the lysis buffer described above supplemented with 300 mM imidazole. The protein was further purified by gel filtration in Tris buffer B. The original, uncropped, and minimally adjusted images is uploaded as S1 Raw Images.

## Statistical analysis

Two-tailed Student's *t* tests were used to compare 2 groups of data. One-way or two-way ANOVA was used to compare significant differences between more than 2 groups. $P < 0.05$ was considered statistically significant; *, **, and *** denote $P < 0.05$, $P < 0.01$, and $P < 0.001$, respectively. Graphing and subsequent analysis were performed via Igor Pro (WaveMetrics), Clampfit (Molecular Devices), GraphPad Prism 8 (GraphPad Software Inc., USA), MATLAB (MathWorks), ImageJ (National Institutes of Health), and Excel (Microsoft, USA). For

behavior analysis and electrophysiology, each recording trace was obtained from a different animal. Unless otherwise specified, the data are presented as the mean ± SEM.

## Supporting information

**S1 Fig. The *cpx-1* mutants exhibit hypersensitivity to aldicarb.** (**A**) The amino acid sequence of CPX-1, indicating the N-terminal domain (NTD), accessory domain (AD), central α-helix (CH), and C-terminal domain (CTD). Mutation sites of *ok1552* (green line), *syb3665* (blue stars), and *syb3584* (brown line) are labeled. (**B**) Aldicarb assay for the indicated strains (1 mM aldicarb). Six trials were conducted with 15 worms in each trial. $N = 6$ independent replicates. All the raw data associated with this figure are available in S1 Data.
(PDF)

**S2 Fig. UNC-31 cells autonomously facilitate excitatory evoked release in cholinergic motor neurons.** (**A**) Representative traces of evoked EPSCs recorded from wild-type, *unc-31 (e928)*, and rescue strains *unc-31(e928); gaaEx1302*. (**B**, **C**) Quantification of the eEPSC amplitude (**B**) and charge transfer (**C**) recorded from the strains mentioned in (**A**). $n \geq 11$ animals. Defects in eEPSCs of *unc-31* mutants can be rescued by expressing UNC-31 cDNA in acetylcholine neurons. The data are presented as box-and-whisker plots, with the median (central line), 25th–75th percentile (bounds of the box), and 5th–95th percentile (whiskers) indicated. One-way ANOVA was used for comparisons of multiple groups ($F_{(2, 35)} = 19.05$, $P < 0.0001$ for **B**; $F_{(2, 35)} = 42.14$, $P < 0.0001$ for **C**), followed by Tukey's range test, * $P < 0.05$; ** $P < 0.01$; *** $P < 0.001$. The error bars represent the SEM. $N = 3$ independent replicates. All the raw data associated with this figure are available in S1 Data.
(PDF)

**S3 Fig. Evoked excitatory neurotransmission regulation by UNC-31 is independent of peptidergic signaling.** (**A**) Representative mPSC traces recorded from WT, *kpc-1(tm1104)*, *aex-5 (sa23)*, and *bli-4(e937)* worms. The right panels show a 0.5 s scale bar for clarity. (**B**, **C**) Quantification of the mPSC frequency (**B**) and amplitude (**C**) across the indicated genotypes, as shown in panel (**A**). $n \geq 10$ animals. One-way ANOVA was performed ($F_{(3, 38)} = 0.1245$, $P = 0.945$ for **B**; $F_{(3, 38)} = 1.652$, $P = 0.1936$ for **C**). (**D**–**F**) Representative traces and quantification of evoked EPSCs recorded from the mentioned genotypes. $n \geq 10$ animals. One-way ANOVA was performed ($F_{(3, 47)} = 0.4829$, $P = 0.6958$ for **E**; $F_{(3, 47)} = 1.201$, $P = 0.3197$ for **F**). The data are presented as box-and-whisker plots, with the median (central line), 25th–75th percentile (bounds of the box), and 5th–95th percentile (whiskers) indicated. One-way ANOVA was performed; ns, not significant. The error bars represent the SEM. $N = 3$ independent replicates. All the raw data associated with this figure are available in S1 Data.
(PDF)

**S4 Fig. Motor impairment in *cpx-1* was exacerbated in *cpx-1; unc-31* double mutants.** (**A**) Motility measured by thrashing number per minute in wild-type, *unc-31(e928)*, *cpx-1(ok1552)*, *cpx-1(ok1552); unc-31(e928)*, *cpx-1(Δ12)*, and *cpx-1(Δ12); unc-31(e928)* worms. $n \geq 39$ animals. (**B**) Quantification of locomotion velocities in each genotype. $n \geq 39$ animals. One-way ANOVA was performed ($F_{(5, 230)} = 673.8$, $P < 0.0001$ for **A**; $F_{(5, 263)} = 566.6$, $P < 0.0001$ for **B**), *** $P < 0.001$. The data are presented as the means ± SEMs. $N = 3$ independent replicates. All the raw data associated with this figure are available in S1 Data.
(PDF)

**S5 Fig. The *unc-31* mutation exacerbates the evoked but not the spontaneous release of inhibitory synaptic transmission in *cpx-1* mutants.** (**A**) Representative traces of eIPSCs

recorded from the indicated strains. The lower traces of *cpx-1(ok1552)* (green) and *cpx-1 (ok1552); unc-31(e928)* (blue) show a zoomed-in region from the upper traces. (**B**, **C**) Quantification of the eIPSC amplitude (**B**) and charge transfer (**C**) at each strain. $n \geq 8$ animals. One-way ANOVA was performed ($F_{(4, 48)} = 33.35$, $P < 0.0001$ for **B**; $F_{(4, 48)} = 21.53$, $P < 0.0001$ for **C**). (**D**) Representative traces of mIPSCs recorded from the indicated strains at different timescales. (**E**, **F**) Quantification of the mIPSC frequency (**E**) and amplitude (**F**) at each strain. $n \geq 11$ animals. One-way ANOVA was performed ($F_{(4, 58)} = 1.786$, $P = 0.1439$ for **E**; $F_{(4, 58)} = 2.579$, $P = 0.0467$ for **F**). The data are presented as box-and-whisker plots, with the median (central line), 25th–75th percentile (bounds of the box), and 5th–95th percentile (whiskers) indicated. Student's *t* test was performed for comparisons of 2 groups, ## $P < 0.01$; ### $P < 0.001$. One-way ANOVA was used for comparisons of multiple groups, followed by Tukey's range test, * $P < 0.05$; ** $P < 0.01$; *** $P < 0.001$; ns, not significant. The error bars represent the SEM. $N = 3$ independent replicates. All the raw data associated with this figure are available in S1 Data.
(PDF)

**S1 Table. *C. elegans* strains used in this study.** Genotypes of experiments in each figure.
(DOCX)

**S2 Table. List of the plasmids used in this study.** Plasmids for the generation of transgenic strains.
(DOCX)

**S3 Table. List of the primers used in this study.** Primers used for PCR amplification for identification of mutants and construction of transgenic strains.
(DOCX)

**S1 Movie. Body curvature and motility in wild-type N2.** Free-moving video of wild-type N2 nematodes. The video is played at 4× speed.
(AVI)

**S2 Movie. Body curvature and motility in *cpx-1(ok1552)* mutants.** Free-moving video of *cpx-1(ok1552)* nematodes. The video is played at 4× speed.
(AVI)

**S3 Movie. Body curvature and motility in *cpx-1(Δ12)* mutants.** Free-moving video of *cpx-1 (Δ12)* nematodes. The video is played at 4× speed.
(AVI)

**S4 Movie. Body curvature and motility in *unc-31(e928)* mutants.** Free-moving video of *unc-31(e928)* nematodes. The video is played at 4× speed.
(AVI)

**S5 Movie. Body curvature and motility in *cpx-1(ok1552); unc-31(e928)* mutants.** Free-moving video of *cpx-1(ok1552); unc-31(e928)* nematodes. The video is played at 4× speed.
(AVI)

**S6 Movie. Body curvature and motility in *cpx-1(Δ12); unc-31(e928)* mutants.** Free-moving video of *cpx-1(Δ12); unc-31(e928)* nematodes. The video is played at 4× speed.
(AVI)

**S1 Raw Images. All the original blot and gel images underlying Fig 5G and 5H.**
(PDF)

**S1 Data. The data underlying the graphs shown in Figs 1, 2, 3, 4, 5, 6, S1, S2, S3, S4, and S5.** (XLSX)

## Acknowledgments

We thank Mei Zhen for reagents, strains, valuable inputs, and comments; Josep Rizo, Shiqing Cai for insightful comments and discussion; and Xia-Jing Tong for strains and helpful suggestions. We thank the *Caenorhabditis Genetics Center*, which is funded by the NIH Office of Research Infrastructure Programs (P40 OD010440, https://orip.nih.gov/), for strains.

## Author Contributions

**Conceptualization:** Shuzo Sugita, Shangbang Gao.

**Data curation:** Ya Wang, Chun Hin Chow, Yu Zhang, Mengjia Huang, Randa Higazy, Xuhui Chen, Yixiang Deng, Sheng Wang.

**Formal analysis:** Ya Wang.

**Funding acquisition:** Shangbang Gao.

**Investigation:** Ya Wang, Chun Hin Chow, Yu Zhang, Mengjia Huang, Randa Higazy, Neeraja Ramakrishnan, Lili Chen, Xuhui Chen, Yixiang Deng, Cong Ma, Shangbang Gao.

**Methodology:** Ya Wang.

**Resources:** Shuzo Sugita.

**Supervision:** Sheng Wang, Cuntai Zhang, Cong Ma, Shuzo Sugita, Shangbang Gao.

**Validation:** Cuntai Zhang, Cong Ma, Shuzo Sugita, Shangbang Gao.

**Writing – original draft:** Shuzo Sugita, Shangbang Gao.

**Writing – review & editing:** Cong Ma, Shuzo Sugita, Shangbang Gao.

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
