## [Editor Report · Decision Letter 0]

17 Oct 2024

Dear Shangbang, 

Thank you for submitting your manuscript entitled "CAPS/UNC-31 Dependence on Complexin/CPX-1 Regulation of Synaptic E/I Balance" for consideration as a Research Article by PLOS Biology.

Your manuscript has now been evaluated by the PLOS Biology editorial staff and I am writing to let you know that we would like to send your submission out for external peer review. Apologies for the delay in getting back to you, but I was unable so far to obtain advice from one of our academic editors. This also means that we have not yet made a firm decision on whether the advance is sufficient for PLOS Biology. We will discuss this aspect with an academic editor once we have received the reviewer reports. 

In any case, before we can send your manuscript to reviewers, we need you to complete your submission by providing the metadata that is required for full assessment. To this end, please login to Editorial Manager where you will find the paper in the 'Submissions Needing Revisions' folder on your homepage. Please click 'Revise Submission' from the Action Links and complete all additional questions in the submission questionnaire.

Once your full submission is complete, your paper will undergo a series of checks in preparation for peer review. After your manuscript has passed the checks it will be sent out for review. To provide the metadata for your submission, please Login to Editorial Manager (https://www.editorialmanager.com/pbiology) within two working days, i.e. by Oct 19 2024 11:59PM.

Kind regards,

Christian

Christian Schnell, PhD

Senior Editor

PLOS Biology

cschnell@plos.org

---

## [Decision Letter · Decision Letter 1]

13 Dec 2024

Dear Shangbang,

Thank you for your patience while we considered your revised manuscript "CAPS/UNC-31 Dependence on Complexin/CPX-1 Regulation of Synaptic E/I Balance" for publication as a Research Article at PLOS Biology. This revised version of your manuscript has been evaluated by the PLOS Biology editors, the Academic Editor and one of the original reviewers.

Based on the reviews and on our Academic Editor's assessment of your revision, we are likely to accept this manuscript for publication, provided you satisfactorily address the following data and other policy-related requests:

* We would like to suggest a different title to improve its accessibility for our broad audience: The exocytosis regulator complexin controls spontaneous synaptic vesicle release in a CAPS-dependent manner at C. elegans inhibitory synapses

* Please add the links to the funding agencies in the Financial Disclosure statement in the manuscript details.

* DATA POLICY:

Regardless of the method selected, please ensure that you provide the individual numerical values that underlie the summary data displayed in the following figure panels as they are essential for readers to assess your analysis and to reproduce it: 1BEF, 2BCEFHIKL, 3EFKLNO, 4BFGHI, 5BCEF, 6BD, Fig3S1BC, Fig3S2BCEF, Fig4S1 and Fig4S2BCEF.

* CODE POLICY

We require the original, uncropped and minimally adjusted images supporting all blot and gel results reported in an article's figures or Supporting Information files. We will require these files before a manuscript can be accepted so please prepare and upload them now. Please carefully read our guidelines for how to prepare and upload this data: https://journals.plos.org/plosbiology/s/figures#loc-blot-and-gel-reporting-requirements

We expect to receive your revised manuscript within two weeks. 

*Published Peer Review History*

*Press*

Sincerely,

Christian

Christian Schnell, PhD

Senior Editor

cschnell@plos.org

PLOS Biology

Reviewer remarks:

Reviewer #1: This manuscript presents robust genetic, physiological, and behavioral evidence highlighting the UNC-31 dependence of CPX-1 in regulating synaptic secretion. These findings provide significant insights into the role of complexin in synaptic function, particularly its contribution to motor behavior and processes such as maintaining the E/I balance.

The revision thoroughly and thoughtfully addresses the technical and conceptual concerns raised by all reviewers. The newly added data are compelling, substantially enhancing the rigor and impact of the study. I think that the manuscript is now suitable for publication in PLOS Biology.

---

## [Editor Report · Decision Letter 2]

16 Jan 2025

Dear Shangbang,

Thank you for the submission of your revised Research Article "The exocytosis regulator complexin controls spontaneous synaptic vesicle release in a CAPS-dependent manner at C. elegans excitatory synapses" for publication in PLOS Biology. Thank you also for clarifying the source data questions. I have uploaded the corrected source data file now. 

On behalf of my colleagues and the Academic Editor, Eunjoon Kim, I am pleased to say that we can in principle accept your manuscript for publication, provided you address any remaining formatting and reporting issues. These will be detailed in an email you should receive within 2-3 business days from our colleagues in the journal operations team; no action is required from you until then. Please note that we will not be able to formally accept your manuscript and schedule it for publication until you have completed any requested changes.

PRESS

Sincerely, 

Christian

Christian Schnell, PhD

Senior Editor

PLOS Biology

cschnell@plos.org